# Acceleration of Federated Learning with Alleviated Forgetting in Local Training

**Chencheng Xu,**[1,2] **Zhiwei Hong,**[1,2] **Minlie Huang,**[1,2*] **Tao Jiang**[3,1,2*]
[1]BNRIST, Tsinghua University, Beijing 100084, China
[2]Department of Computer Science and Technology, Tsinghua University, Beijing 100084, China
[3]Department of Computer Science and Engineering, UCR, CA 92521, USA
`{xucc18, hzw17}@mails.tsinghua.edu.cn`
`aihuang@tsinghua.edu.cn, jiang@cs.ucr.edu`

## Abstract

Federated learning (FL) enables distributed optimization of machine learning models while protecting privacy by independently training local models on each client and then aggregating parameters on a central server, thereby producing an effective global model. Although a variety of FL algorithms have been proposed, their training efficiency remains low when the data are not independently and identically distributed (non-i.i.d.) across different clients. We observe that the slow convergence rates of the existing methods are (at least partially) caused by the catastrophic forgetting issue during the local training stage on each individual client, which leads to a large increase in the loss function concerning the previous training data at the other clients. Here, we propose FedReg, an algorithm to accelerate FL with alleviated knowledge forgetting in the local training stage by regularizing locally trained parameters with the loss on generated pseudo data, which encode the knowledge of previous training data learned by the global model. Our comprehensive experiments demonstrate that FedReg not only significantly improves the convergence rate of FL, especially when the neural network architecture is deep and the clients' data are extremely non-i.i.d., but is also able to protect privacy better in classification problems and more robust against gradient inversion attacks. The code is available at: `https://github.com/Zoesgithub/FedReg`.

## 1 Introduction

Federated learning (FL) has emerged as a paradigm to train a global machine learning model in a distributed manner while taking privacy concerns and data protection regulations into consideration by keeping data on clients (Voigt & Von dem Bussche, 2017). The main challenge faced in FL is how to reduce the communication costs (in training) without degrading the performance of the final resultant model, especially when the data on different clients are not independently and identically distributed (non-i.i.d.) (Yuan & Ma, 2020; Wang et al., 2020b). The most popular FL algorithm is FedAvg (McMahan et al., 2017a). In each training round of FedAvg, local training steps are separately performed at every client and the locally trained models are transferred to the server. Then, the server aggregates the local models into a global model by simply averaging their parameters. Although FedAvg succeeds in many applications, its training processes often diverge when the data are non-i.i.d., also known as heterogeneous, across the clients (Li et al., 2019; Zhao et al., 2018). Several FL algorithms have been designed to improve FedAvg and tackle the heterogeneity issue mainly by reducing the difference between locally trained parameters (Li et al., 2018; Karimireddy et al., 2020) or aggregating these parameters into different groups (Wang et al., 2020a; Yurochkin et al., 2019). However, the performance of these methods is still far from satisfactory when a deep neural network architecture is employed (Rothchild et al., 2020; Li et al., 2021). On the other hand, recent work in the literature (Geiping et al., 2020) shows that the transmission of trained model parameters does not ensure the protection of privacy. Privacy-sensitive information can be recovered

---

[*]Minlie Huang and Tao Jiang are the co-corresponding authors.

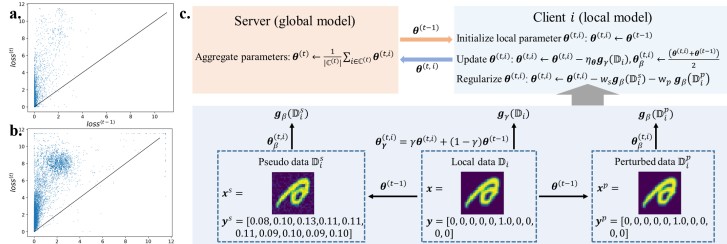

Figure 1: The catastrophic forgetting issues on non-i.i.d. MNIST (a) and EMNIST (b) data, and an illustration of FedReg (c). Here, $loss^{(t-1)}$ denotes the loss on data $\mathbb{D}_j$, which is the local data of client $j$ sampled in round $t-1$, computed with parameters $\boldsymbol{\theta}^{(t-1)}$, and $loss^{(t)}$ denotes the averaged loss on the same data computed with the locally trained parameters in round $t$. See Appendix B.1 for their detailed definitions. The values of $loss^{(t)}$ are significantly larger than those of $loss^{(t-1)}$, indicating that after the local training stage, the knowledge about the training data in the previous round at the other clients has been forgotten. To accelerate the convergence rate of FL by alleviating such forgetting issue in the local training stage, FedReg generates pseudo data $\mathbb{D}_i^s$ and perturbed data $\mathbb{D}_i^p$, and regularizes the local parameter $\boldsymbol{\theta}^{(t,i)}$ by constraining the loss on $\mathbb{D}_i^s$ and $\mathbb{D}_i^p$. Therefore, instead of using the plain gradient with learning rate $\eta_{\boldsymbol{\theta}}$, FedReg updates $\boldsymbol{\theta}^{(t,i)}$ with the gradient $\boldsymbol{g}_\gamma(\mathbb{D}_i)$ computed with slowly-updated parameters $\boldsymbol{\theta}_\gamma^{(t,i)}$, which prevents the gradient from converging to zero in few steps, and then it regularizes $\boldsymbol{\theta}^{(t,i)}$ with a combination of the gradients $\boldsymbol{g}_\beta(\mathbb{D}_i^s)$ and $\boldsymbol{g}_\beta(\mathbb{D}_i^p)$ computed from the pseudo and perturbed data.

with gradient inversion attacks (Zhu et al., 2019). Differential privacy (DP) (McMahan et al., 2017b; Abadi et al., 2016) is one of the most widely used strategies to prevent the leakage of private information. However, when DP is incorporated into FL, the performance of the resulting model decays significantly (Jayaraman & Evans, 2019).

We observe that when the data are non-i.i.d. across the clients, the locally trained models suffer from severe forgetting of the knowledge of previous training data at the other clients (*i.e.*, the well-known catastrophic forgetting issue), perhaps due to the discrepancy between local data distributions and the global data distribution. As shown in Figure 1 and supplementary Figure C.1, this forgetting issue leads to a large increase in the loss concerning these training data under the non-i.i.d. setting, thereby slowing down the convergence speed. In this work, we propose a novel algorithm, FedReg, that reduces the communication costs in training by alleviating the catastrophic forgetting issue in the local training stage. FedReg reduces knowledge forgetting by *regularizing* the locally trained parameters with generated pseudo data, which are obtained by using modified local data to encode the knowledge of previous training data as learned by the global model. The potential conflict with the knowledge in the local data introduced by the pseudo data is dampened by the use of perturbed data, which are generated by making small perturbations to the local data, whose predictive values they help ensure. The generation of the pseudo data and perturbed data only relies on the global model received from the server and the local data on the current client. Thus, compared with FedAvg, no extra communication costs concerning the data on the other clients are incurred. Our extensive experiments demonstrate the superiority of FedReg in accelerating the FL training process, especially when the neural network architecture is deep and the clients' data are extremely heterogeneous. Furthermore, the pseudo data can be further utilized in classification problems to defend against gradient inversion attacks. More specifically, we show that with similar degree of protection of private information, the degradation in the performance of the global model learned by our method is much less than that learned by using FedAvg combined with DP.

**Our contributions.** We demonstrate that when the data across clients are non-i.i.d., catastrophic forgetting in the local training stage is an important factor that slows down the FL training process. We therefore propose a novel algorithm, FedReg, that accelerates FL by alleviating catastrophic forgetting with generated pseudo data. We also perform extensive experiments to establish the superiority of FedReg. We further show that in classification problems, the pseudo data generated in FedReg can help protect private information from gradient inversion attacks with much less impact on the performance of the resultant model compared with DP.

## 2 RELATED WORK

### 2.1 FEDERATED LEARNING

FL is proposed to address privacy concerns in a distributed learning environment. In the FL paradigm, a client $i$ keeps its local data $\mathbb{D}_i$ on the client machine during the whole training process so that the server and other clients cannot directly access $\mathbb{D}_i$. The objective is to find the parameter $\boldsymbol{\theta}^*$ that minimizes the loss on the global data $\cup_{i\in\mathbb{C}}\mathbb{D}_i$, where $\mathbb{C}$ is the set of clients, *i.e.*,

$$\boldsymbol{\theta}^* = \arg\min_{\boldsymbol{\theta}} \frac{1}{\sum_{i\in\mathbb{C}}|\mathbb{D}_i|} \sum_{\boldsymbol{d}\in\cup_{i\in\mathbb{C}}\mathbb{D}_i} L_{\boldsymbol{\theta}}\left(\boldsymbol{d}\right), \tag{1}$$

In this equation, $L_{\boldsymbol{\theta}}$ is the loss function with parameters $\boldsymbol{\theta}$, which can be cross-entropy or in some other custom-defined form. FedAvg (McMahan et al., 2017a) and FedProx (Li et al., 2018) are the most widely used FL algorithms. In FedAvg, in a training round $t$, the server sends the initial parameters $\boldsymbol{\theta}^{(t-1)}$ to a set of sampled clients $\mathbb{C}^{(t)}$. Then, the parameters are updated independently for $S$ epochs on each of these clients to minimize the loss on the local data, and the locally trained parameters $\boldsymbol{\theta}^{(t,i)}$ are then sent to the server. The server aggregates the parameters by simply averaging over them and obtains the parameters $\boldsymbol{\theta}^{(t)}$, *i.e.*, $\boldsymbol{\theta}^{(t)} = \frac{1}{K}\sum_{i\in\mathbb{C}^{(t)}}\boldsymbol{\theta}^{(t,i)}$, where $K$ is the number of sampled clients in round $t$. FedAvg is unstable and diverge when the data are non-i.i.d. across different clients (Li et al., 2018). FedProx stabilizes FedAvg by including a proximal term in the loss function to limit the distance between $\boldsymbol{\theta}^{(t,i)}$ and $\boldsymbol{\theta}^{(t-1)}$. Although FedProx provides a theoretical proof of convergence, empirically it fails to achieve good performances when deep neural network architectures are used (Li et al., 2021). SCAFFOLD (Karimireddy et al., 2020) assumes that the heterogeneity of data leads to a client-drift in gradients and correlates the drift with gradient correction terms. As clients need to send extra information to the server, SCAFFOLD increases the risk of privacy leakage and doubles the communication burden compared with FedAvg. Furthermore, the accuracy of the correction terms is highly correlated with the training history of the client. Thus, the performance of SCAFFOLD decreases significantly when the number of clients is large, in which case each client is only sampled for few times and the estimation of the gradient correction terms is often inaccurate. FedCurv (Shoham et al., 2019) aims to tackle the forgetting issue on non-i.i.d. data with elastic weight consolidation (Kirkpatrick et al., 2017). To achieve this, FedCurv needs to transfer Fisher information between the server and clients, which increases the communication costs to $2.5$ times compared with FedAvg. Multiple methods (Luo et al., 2021; Hao et al., 2021; Goetz & Tewari, 2020) have also tried to introduce synthesized data to help reduce the effect of heterogeneity, but they either rely on some specific architectures of neural networks (Luo et al., 2021) such as batch normalization (Ioffe & Szegedy, 2015) or require the synthesized data to be shared with the server (Hao et al., 2021; Goetz & Tewari, 2020), which contradicts the privacy protection objectives of FL.

### 2.2 CATASTROPHIC FORGETTING

Catastrophic forgetting occurs specifically when the neural network is trained sequentially on multiple tasks. In this case, the optimal parameters for the current task might perform poorly on the objectives of previous tasks. Many algorithms have been proposed to alleviate the forgetting issue. Memory-based methods (Parisi et al., 2019) have achieved excellent performances in accommodating new knowledge while retaining previously learned experience. Such memory-based methods, such as gradient episodic memory (GEM) (Lopez-Paz & Ranzato, 2017) and averaged gradient episodic memory (A-GEM) (Chaudhry et al., 2018), store a subset of data from previous tasks and replay the memorized data when training on the current task. For instance, A-GEM treats the losses on the episodic memories of previous tasks as inequality constraints in optimizing the objectives of current tasks and changes the model updates from the plain gradient $\boldsymbol{g}$ to $\boldsymbol{g} - w\boldsymbol{g}_{ref}$, where $\boldsymbol{g}_{ref}$ is the gradient computed from the loss on the memorized data and $w$ is a non-negative weight. Unfortunately, these memory-based techniques are not suitable for FL due to privacy concerns.

### 2.3 GRADIENT INVERSION ATTACKS

FL provides a privacy guarantee by keeping the users' data on individual client machines locally and only sharing the model updates. However, recent research has shown that data information can be recovered from the parameter updates (Geiping et al., 2020; Zhu et al., 2019) in the FedAvg

framework by simply finding data with updates similar to the values returned from the client. DP, which avoids privacy leakage by adding noise to training data (Sun et al., 2019) or model updates (Abadi et al., 2016; McMahan et al., 2017b), is the most widely used strategy to protect private information. When adding noise to model updates, such as differentially private SGD (DPSGD) (Abadi et al., 2016), the noise level in DP is controlled by the gradient norm bound $C$ and the noise scale $\sigma$. DP often causes a large performance degradation in the resultant model.

## 3 METHOD

Our main challenge is how to alleviate the forgetting of previously learned knowledge at each client without having to access data at the other clients in the local training stage. For any data set $\mathbb{D}$, we denote the set of data near $\mathbb{D}$ within the Euclidean distance of $\delta$ as $\mathbb{N}(\mathbb{D}, \delta) = \cup_{\boldsymbol{d}=(\boldsymbol{x},\boldsymbol{y})\in\mathbb{D}}\{(\boldsymbol{x}',\boldsymbol{y}')\,|\,0 < \|\boldsymbol{x} - \boldsymbol{x}'\| \leq \delta\}$. We assume that in round $t$ on client $i$, (1) for all data $\boldsymbol{d}^- \in \cup_{j\in\mathbb{C}/\{i\}}\mathbb{D}_j$ local to the other clients, the global model with parameter $\boldsymbol{\theta}^{(t-1)}$ has a better feature representation than the local model with parameter $\boldsymbol{\theta}^{(t,i)}$; and (2) for any $c > 0$, $\exists \epsilon, \delta > 0$ such that if $\forall \boldsymbol{d}' = (\boldsymbol{x}',\boldsymbol{y}') \in \mathbb{N}(\mathbb{D}_i, \delta)$, $L_{\boldsymbol{\theta}^{(t,i)}}((\boldsymbol{x}',\boldsymbol{f}_{\boldsymbol{\theta}^{(t-1)}}(\boldsymbol{x}'))) - L_{\boldsymbol{\theta}^{(t-1)}}((\boldsymbol{x}',\boldsymbol{f}_{\boldsymbol{\theta}^{(t-1)}}(\boldsymbol{x}'))) \leq \epsilon$, then $\forall \boldsymbol{d}^- = (\boldsymbol{x}^-,\boldsymbol{y}^-) \in \cup_{j\in\mathbb{C}/\{i\}}\mathbb{D}_j$, $L_{\boldsymbol{\theta}^{(t,i)}}((\boldsymbol{x}^-,\boldsymbol{f}_{\boldsymbol{\theta}^{(t-1)}}(\boldsymbol{x}^-))) - L_{\boldsymbol{\theta}^{(t-1)}}((\boldsymbol{x}^-,\boldsymbol{f}_{\boldsymbol{\theta}^{(t-1)}}(\boldsymbol{x}^-))) \leq c$, where $\boldsymbol{f}_{\boldsymbol{\theta}}$ is the prediction function with parameters $\boldsymbol{\theta}$. The assumption (2) guarantees that, in the local training stage, the range of changes in the predicted values of previous training data at the other clients can be controlled by constraining the changes in the predicted values of the data near $\mathbb{D}_i$. Based on these assumptions, we first generate pseudo data and then alleviate the catastrophic forgetting issue by regularizing the locally trained parameters with the loss on the pseudo data. In other words, we hope that the pseudo data would achieve the same effect as the previous training data in constraining the local model. Note that FedReg does not assume any particular form of the loss function, but the method of modifying gradients to enhance privacy protection (to be discussed below in Section 3.3) is currently only applicable to classification problems. The pseudo code of FedReg is provided in Appendix D.

### 3.1 GENERATION OF PSEUDO DATA AND PERTURBED DATA

With the above assumptions, the catastrophic forgetting issue can be alleviated by constraining the prediction values for data in $\mathbb{N}(\mathbb{D}, \delta)$. However, such a constraint is too strong and would hinder the learning of new knowledge from the local data. Moreover, it is also computationally inefficient to enumerate all such data. Observe that for a data point $\boldsymbol{d}' = (\boldsymbol{x}',\boldsymbol{y}') \in \mathbb{N}(\mathbb{D}_i, \delta)$, after locally training $\boldsymbol{\theta}^{(t,i)}$ with $\mathbb{D}_i$, if $\boldsymbol{f}_{\boldsymbol{\theta}^{(t,i)}}(\boldsymbol{x}') = \boldsymbol{f}_{\boldsymbol{\theta}^{(t-1)}}(\boldsymbol{x}')$ then the constraint on $\boldsymbol{d}'$ will not contribute to addressing the forgetting issue. Therefore, to efficiently regularize the locally trained parameters, only data points with large changes in the predicted values should be used to alleviate knowledge forgetting. Such data points are usually within a small distance to the data points in $\mathbb{D}_i$, but their predicted values given by $\boldsymbol{\theta}^{(t-1)}$ could be very different from the labels in $\mathbb{D}_i$. Data satisfying these conditions can be easily obtained with the fast gradient sign method (Goodfellow et al., 2014) and we denote the data generated in this way as pseudo data $\mathbb{D}_i^s$. More precisely, each data point $(\boldsymbol{x}^s, \boldsymbol{y}^s) \in \mathbb{D}_i^s$ is iteratively generated from a data point $(\boldsymbol{x}, \boldsymbol{y}) \in \mathbb{D}_i$ as follows:

$$\boldsymbol{x}^s = \boldsymbol{x}_E^s, \; \boldsymbol{y}^s = \boldsymbol{f}_{\boldsymbol{\theta}^{(t-1)}}(\boldsymbol{x}^s), \tag{2}$$

$$\boldsymbol{x}_j^s = \boldsymbol{x}_{j-1}^s + \eta_s \mathrm{sign}\left(\nabla_{\boldsymbol{x}_{j-1}^s} L_{\boldsymbol{\theta}^{(t-1)}}\left((\boldsymbol{x}_{j-1}^s, \boldsymbol{y})\right)\right), \; j = 1, ..., E, \; \boldsymbol{x}_0^s = \boldsymbol{x},$$

where $\eta_s$ is the step size and $E$ the number of steps. Despite the relaxation in constraints achieved by the pseudo data generated above, some information conflicting with the knowledge embedded in the local data $\mathbb{D}_i$ could possibly be introduced in $\mathbb{D}_i^s$ due to the inaccuracy of the global model during the training process. To further eliminate such conflicting information, perturbed data $\mathbb{D}_i^p = \{(\boldsymbol{x}^p, \boldsymbol{y}^p)\}$ with small perturbations on $\mathbb{D}_i$ are iteratively generated as follows:

$$\boldsymbol{x}^p = \boldsymbol{x}_E^p, \; \boldsymbol{y}^p = \boldsymbol{y}, \; \boldsymbol{x}_j^p = \boldsymbol{x}_{j-1}^p + \eta_p \mathrm{sign}\left(\nabla_{\boldsymbol{x}_{j-1}^p} L_{\boldsymbol{\theta}^{(t-1)}}\left((\boldsymbol{x}_{j-1}^p, \boldsymbol{y})\right)\right), \; j = 1, ..., E, \; \boldsymbol{x}_0^p = \boldsymbol{x}, \tag{3}$$

where the step size $\eta_p$ satisfies $\eta_p \ll \eta_s$ in order to make sure that the perturbed data is much closer to the local data than the pseudo data. In our experiments, we take $\eta_p = 0.01\eta_s$ and $E = 10$ to reduce the complexity of tuning hyper-parameters.

## 3.2 ALLEVIATION OF CATASTROPHIC FORGETTING

With the pseudo data $\mathbb{D}_i^s$ and perturbed data $\mathbb{D}_i^p$ generated above, we regularize $\boldsymbol{\theta}^{(t,i)}$ by requiring

$$\sum_{\boldsymbol{d}^s \in \mathbb{D}_i^s} L_{\boldsymbol{\theta}^{(t,i)}}\left(\boldsymbol{d}^s\right) \le \sum_{\boldsymbol{d}^s \in \mathbb{D}_i^s} L_{\boldsymbol{\theta}^{(t-1)}}\left(\boldsymbol{d}^s\right), \tag{4}$$

$$\sum_{\boldsymbol{d}^p \in \mathbb{D}_i^p} L_{\boldsymbol{\theta}^{(t,i)}}\left(\boldsymbol{d}^p\right) \le \sum_{\boldsymbol{d}^p \in \mathbb{D}_i^p} L_{\boldsymbol{\theta}^{(t-1)}}\left(\boldsymbol{d}^p\right), \tag{5}$$

where constraint (4) alleviates the catastrophic forgetting issue and constraint (5) eliminates conflicting information introduced in (4). Constraint (5) also helps improve the robustness of the resultant model (Madry et al., 2018). Expanding both sides of (4) and of (5) respectively with $\boldsymbol{\theta}_\beta^{(t,i)} = 0.5\left(\boldsymbol{\theta}^{(t,i)} + \boldsymbol{\theta}^{(t-1)}\right)$ so that the second-order term can be eliminated and ignoring higher-order terms (see Appendix A.1 for the details), we obtain

$$\left(\boldsymbol{\theta}^{(t-1)} - \boldsymbol{\theta}^{(t,i)}\right)^T \boldsymbol{g}_\beta\left(\mathbb{D}_i^s\right) \ge 0, \ \boldsymbol{g}_\beta\left(\mathbb{D}_i^s\right) = \nabla_{\boldsymbol{\theta}_\beta^{(t,i)}} \frac{1}{|\mathbb{D}_i^s|} \sum_{\boldsymbol{d}^s \in \mathbb{D}_i^s} L_{\boldsymbol{\theta}_\beta^{(t,i)}}\left(\boldsymbol{d}^s\right) \tag{6}$$

$$\left(\boldsymbol{\theta}^{(t-1)} - \boldsymbol{\theta}^{(t,i)}\right)^T \boldsymbol{g}_\beta\left(\mathbb{D}_i^p\right) \ge 0, \ \boldsymbol{g}_\beta\left(\mathbb{D}_i^p\right) = \nabla_{\boldsymbol{\theta}_\beta^{(t,i)}} \frac{1}{|\mathbb{D}_i^p|} \sum_{\boldsymbol{d}^p \in \mathbb{D}_i^p} L_{\boldsymbol{\theta}_\beta^{(t,i)}}\left(\boldsymbol{d}^p\right) \tag{7}$$

Due to the existence of potentially conflicting information in $\mathbb{D}_i^s$ and $\mathbb{D}_i^p$, directly finding $\boldsymbol{\theta}^{(t,i)}$ to satisfy the above two inequalities is mathematically ill-defined. Besides, given the complex architectures of deep neural networks, it is computationally too challenging to solve either of the above inequalities. Therefore, we approximate the optimal parameters $\boldsymbol{\theta}^{(t,i)^*}$ by solving the following constrained optimization problems:

$$\boldsymbol{\theta}^{(t,i)'} = \arg\min_{\boldsymbol{\theta}} \|\boldsymbol{\theta} - \boldsymbol{\theta}^{(t,i)}\|^2 \ s.t. \ \left(\boldsymbol{\theta}^{(t-1)} - \boldsymbol{\theta}\right)^T \boldsymbol{g}_\beta\left(\mathbb{D}_i^s\right) \ge 0, \tag{8}$$

$$\boldsymbol{\theta}^{(t,i)^*} = \arg\min_{\boldsymbol{\theta}} \|\boldsymbol{\theta} - \boldsymbol{\theta}^{(t,i)'}\|^2 \ s.t. \ \left(\boldsymbol{\theta}^{(t-1)} - \boldsymbol{\theta}\right)^T \boldsymbol{g}_\beta(\mathbb{D}_i^p) \ge 0. \tag{9}$$

By solving (8) and (9) (see Appendix A.2 for the detailed derivation), we obtain

$$\boldsymbol{\theta}^{(t,i)^*} = \boldsymbol{\theta}^{(t,i)} - w_s \boldsymbol{g}_\beta\left(\mathbb{D}_i^s\right) - w_p \boldsymbol{g}_\beta(\mathbb{D}_i^p), \tag{10}$$

$$w_s = \max\left(\frac{\left(\boldsymbol{\theta}^{(t,i)} - \boldsymbol{\theta}^{(t-1)}\right)^T \boldsymbol{g}_\beta(\mathbb{D}_i^s)}{\boldsymbol{g}_\beta\left(\mathbb{D}_i^s\right)^T \boldsymbol{g}_\beta\left(\mathbb{D}_i^s\right)}, 0\right), \ w_p = \max\left(\frac{\left(\boldsymbol{\theta}^{(t,i)} - w_s \boldsymbol{g}_\beta(\mathbb{D}_i^s) - \boldsymbol{\theta}^{(t-1)}\right)^T \boldsymbol{g}_\beta(\mathbb{D}_i^p)}{\boldsymbol{g}_\beta\left(\mathbb{D}_i^p\right)^T \boldsymbol{g}_\beta\left(\mathbb{D}_i^p\right)}, 0\right).$$

Based on the generation process of the pseudo data, it is easy to note that $\sum_{\boldsymbol{d}^s \in \mathbb{D}_i^s} L_{\boldsymbol{\theta}^{(t-1)}}(\boldsymbol{d}^s) = \min_{\boldsymbol{\theta}} \sum_{\boldsymbol{d}^s \in \mathbb{D}_i^s} L_{\boldsymbol{\theta}}(\boldsymbol{d}^s)$, and thus the inequality sign in (4) can in fact be replaced by an equality sign. In practice, however, we have observed that the training process would be more stable and result in non-negative $w_s$ and $w_p$ with the inequality formulation, probably due to errors introduced in the above approximation. Finally, to prevent the gradient from converging to zero in a very small number of steps, slowly-updated parameters (Zhang et al., 2019b) $\boldsymbol{\theta}_\gamma^{(t,i)} = \gamma \boldsymbol{\theta}^{(t,i)} + (1-\gamma)\boldsymbol{\theta}^{(t-1)}$ with $\gamma \in [0,1]$ are used to compute the gradient for $\mathbb{D}_i$, *i.e.*, $\boldsymbol{g}_\gamma(\mathbb{D}_i) = \nabla_{\boldsymbol{\theta}_\gamma^{(t,i)}} \frac{1}{|\mathbb{D}_i|} \sum_{\boldsymbol{d} \in \mathbb{D}_i} L_{\boldsymbol{\theta}_\gamma^{(t,i)}}(\boldsymbol{d})$.

In summary, taking into acount the above improvements, the local parameters are updated in each training step as:

$$\boldsymbol{\theta}^{(t,i)} \leftarrow \boldsymbol{\theta}^{(t,i)} - \eta_{\boldsymbol{\theta}} \boldsymbol{g}_\gamma\left(\mathbb{D}_i\right), \ \boldsymbol{\theta}^{(t,i)} \leftarrow \boldsymbol{\theta}^{(t,i)} - w_s \boldsymbol{g}_\beta\left(\mathbb{D}_i^s\right) - w_p \boldsymbol{g}_\beta\left(\mathbb{D}_i^p\right). \tag{11}$$

## 3.3 MODIFICATION OF GRADIENTS TO PROTECT PRIVACY

We further notice that in classification problems, the pseudo data can be used to modify the gradient $\boldsymbol{g}_\gamma(\mathbb{D}_i)$ to enhance the protection of privacy. If we denote $\mathbb{D}_i^{s'} = \{(\boldsymbol{x}, \frac{1}{n}\sum_{j=1}^n \boldsymbol{e}^{(j)}) | (\boldsymbol{x}, \boldsymbol{y}) \in \mathbb{D}_i^s\}$ for an $n$-classification problem, where $\boldsymbol{e}^{(j)}$ is the standard basis vector with a 1 at position $j$, then the data points in $\mathbb{D}_i^{s'}$ are similar to those in $\mathbb{D}_i$, but they may have totally different labels. Hence,

it is reasonable to assume that $\boldsymbol{g}_\gamma(\mathbb{D}_i)$ and $\boldsymbol{g}_\gamma(\mathbb{D}_i^{s'})$ contain similar semantic information but different classification information. Since in the training of FL, only the classification information contributes to the model performance, removing the semantic information in $\boldsymbol{g}_\gamma(\mathbb{D}_i)$ will enhance the privacy protection capability of FL without severely degrading the performance of the resultant model. Based on this intuition, we presume that the components of $\boldsymbol{g}_\gamma(\mathbb{D}_i)$ in the same (or roughly the same) direction of $\boldsymbol{g}_\gamma(\mathbb{D}_i^{s'})$ contain the semantic information and the other (orthogonal) components contain the classification information. Thus, we compute a modified gradient (MG) to enhance the protection of privacy as

$$\tilde{\boldsymbol{g}}_\gamma\left(\mathbb{D}_i\right) = \boldsymbol{g}_\gamma\left(\mathbb{D}_i\right) - v\boldsymbol{g}_\gamma\left(\mathbb{D}_i^{s'}\right), \ v = \frac{\boldsymbol{g}_\gamma\left(\mathbb{D}_i\right)^T \boldsymbol{g}_\gamma\left(\mathbb{D}_i^{s'}\right)}{\boldsymbol{g}_\gamma\left(\mathbb{D}_i^{s'}\right)^T \boldsymbol{g}_\gamma\left(\mathbb{D}_i^{s'}\right)} \tag{12}$$

We refer to the version of FedReg in which the $\boldsymbol{g}_\gamma\left(\mathbb{D}_i\right)$ term in (11) is replaced by $\tilde{\boldsymbol{g}}_\gamma\left(\mathbb{D}_i\right)$ as FedReg with MG.

## 4 EXPERIMENTS

We compare the performance of FedFeg against the above-mentioned baseline FL algorithms including FedAvg, FedProx, FedCurv and SCAFFOLD as well as the well-known SGD algorithm (Bottou, 2012). Note that SGD corresponds to a special case of FedAvg where the local training stage consists of only one step and all data on a client are used in a large single batch.

### 4.1 DATA PREPARATION

FedReg is evaluated on MNIST (Deng, 2012), EMNIST (Cohen et al., 2017), CIFAR-10 and CIFAR-100 (Krizhevsky et al., 2009), and CT images of COVID-19 (He, 2020). To simulate a scenario for FL, the training data in each dataset are split into multiple clients in different ways and the performance of the trained model is evaluated on the test data. The data preparation steps for each dataset are described below and more experimental details are provided in Appendix B.

**MNIST** The training data are split into 5,000 clients. Each client has images either from only one class, named MNIST (one class), or from two classes, named MNIST (two classes). The number of images per client follows a power law distribution (Li et al., 2018). A 5-layer neural network with three layers of convolutional neural networks (CNN) and two layers of full connections (FC) is used.

**EMNIST** The training data of the digits in EMNIST are split into 10,000 clients. Each client owns 24 images from the same class. The same model architecture used on MNIST is used here.

**CIFAR-10** The training data are split into 10,000 clients. Each client owns 5 images from random classes, named CIFAR-10 (uniform), or from only one class, named CIFAR-10 (one class). A ResNet-9 network with the Fixup initialization (Zhang et al., 2019a) is trained from scratch.

**CIFAR-100** The training data are split into 50,000 clients and each client owns 1 image. A ResNet-9 network with the Fixup initialization is trained from scratch, named CIFAR-100 (ResNet-9), and a pre-trained transformer (Dosovitskiy et al., 2020) is fine-tuned, named CIFAR-100 (Transformer).

**CT images related to COVID-19** The dataset contains 349 CT images of COVID-19 patients with source information and 397 non-COVID-19 CT images without source information. The data are split into training and test data in the same way provided in (He, 2020). Then, the CT images of COVID-19 patients in the training data are split to make sure that the images from the same source are assigned to the same client, and 32 clients are obtained in this way. The non-COVID-19 images in the training data are distributed to each client following the proportion of COVID-19 images. A DenseNet-121 network (Huang et al., 2017) and a 10-layer neural network with 9 layers of CNNs and one layer of FC are trained on these CT images to distinguish the COVID-19 images from the non-COVID-19 ones.

### 4.2 COMPARISON OF CONVERGENCE RATES

The results on MNIST and EMNIST are shown in Table 1. FedReg required fewer communication rounds to converge compared with the baseline methods and achieved a higher final accuracy (*i.e.*,

Table 1: Comparison results on MNIST and EMNIST. The ACC columns show the final accuracy on test data. The $R_a$ columns show the minimum number of rounds required to reach $a * 100\%$ of the accuracy of SGD in the corresponding experiment.

| Method | MNIST (two classes) | | | | MNIST (one class) | | | | EMNIST | | | |
|---|---|---|---|---|---|---|---|---|---|---|---|---|
| | $R_{0.5}$ | $R_{0.9}$ | $R_{1.0}$ | ACC | $R_{0.5}$ | $R_{0.9}$ | $R_{1.0}$ | ACC | $R_{0.5}$ | $R_{0.9}$ | $R_{1.0}$ | ACC |
| SGD | 22 | 81 | 478 | 0.975 | 33 | 95 | 422 | 0.976 | 74 | 160 | 470 | 0.982 |
| FedAvg | 5 | 37 | 491 | 0.975 | 28 | 74 | - | 0.976 | 118 | 170 | 466 | 0.984 |
| FedProx | 7 | 41 | - | 0.974 | 21 | 74 | - | 0.975 | 126 | 178 | 446 | 0.984 |
| SCAFFOLD | 7 | 47 | - | 0.957 | 29 | 73 | - | 0.930 | 149 | 221 | - | 0.967 |
| FedCurv | 7 | 43 | - | 0.974 | 29 | 74 | - | 0.976 | 117 | 170 | 432 | 0.985 |
| FedReg | **4** | **25** | **367** | **0.977** | **5** | **32** | **384** | **0.978** | **6** | **24** | **299** | **0.987** |

Table 2: Comparison results on CIFAR-10.

| Method | CIFAR-10 (uniform) | | | | CIFAR-10 (one class) | | | |
|---|---|---|---|---|---|---|---|---|
| | $R_{0.5}$ | $R_{0.9}$ | $R_{1.0}$ | ACC | $R_{0.5}$ | $R_{0.9}$ | $R_{1.0}$ | ACC |
| SGD | 40 | 160 | 185 | 0.483 | 39 | 168 | 197 | 0.449 |
| FedAvg | 4 | 52 | 94 | 0.576 | 108 | - | - | 0.371 |
| FedProx | 6 | 53 | 94 | 0.576 | 108 | - | - | 0.373 |
| SCAFFOLD | 7 | 116 | - | 0.447 | 88 | - | - | 0.230 |
| FedCurv | 4 | 53 | 94 | 0.569 | 108 | - | - | 0.368 |
| FedReg | **2** | **31** | **40** | **0.715** | **9** | **58** | **84** | **0.616** |

accuracy of the respective models after finishing all training rounds) under all the experimental settings, especially on EMNIST, where the number of clients is twice compared to MNIST, showing clearly the superiority of FedReg in dealing with non-i.i.d. data when a shallow model is used and its robust scalability. FedCurv also aims to allieviate the forgetting issue, but its performance is not significantly better than FedProx. Note that due to its heavy reliance on the optimization history when estimating the correction term, in our experiments, SCAFFOLD performed worse than FedAvg. Table 2 illustrates that, on CIFAR-10, when the data are randomly distributed, FedReg significantly outperformed all the baseline methods in both convergence speed (measured in communication rounds) and final model accuracy, demonstrating the superiority of FedReg in training deep neural networks. When data across clients become more heterogeneous and the images at each client are from the same class, the multiple local training steps in the baseline algorithms (other than SGD) failed to improve the performance of the locally trained model, resulting in a worse performance than that of SGD, while FedReg was still able to save about a half of the communication costs compared with SGD. A similar conclusion can be drawn when training the ResNet-9 network from scratch and fine-tuning the transformer model on CIFAR-100, as shown in Table 3, indicating the robustness of FedReg on various datasets and model architectures. The results in distinguishing COVID-19 CT images from non-COVID-19 images on a DenseNet-121 network, as shown in Table 3, further exhibits the superiority of FedReg in medical image processing applications when the model architecture is deep and complex. To assess the advantage of FedReg on real-world data, it is compared with the other methods on a popular real-world dataset Landmarks-User-160k (Hsu et al., 2020; Lai et al., 2021), where the data at each client consists of many classes, and as shown in supplementary Table C.1, FedReg still outperformed the baseline methods. The wall-clock time usages of the methods in each local training round on MNIST and EMNIST are given in supplementary Table C.2. Note that since FedReg performs additional computation in each round, its local training requires a bit more time.

## 4.3 ALLEVIATION OF CATASTROPHIC FORGETTING

To prove that FedReg indeed alleviates the catastrophic forgetting, increases of the loss values concerning previous training data at the other clients are compared between FedReg and FedAvg. As

Table 3: Comparison results on CIFAR-100 and CT images related to COVID-19.

| Method | CIFAR-100 (ResNet-9) | | | | CIFAR-100 (Transformer) | | | | COVID-19 CT Images | |
| --- | --- | --- | --- | --- | --- | --- | --- | --- | --- | --- |
| | $R_{0.5}$ | $R_{0.9}$ | $R_{1.0}$ | ACC | $R_{0.5}$ | $R_{0.9}$ | $R_{1.0}$ | ACC | $R_{1.0}$ | ACC |
| SGD | 510 | 1004 | 1200 | 0.434 | 4 | 16 | 74 | 0.888 | 4 | 0.511 |
| FedAvg | 793 | - | - | 0.323 | 3 | 21 | - | 0.883 | 3 | 0.614 |
| FedProx | 793 | - | - | 0.325 | 3 | 26 | - | 0.880 | 4 | 0.620 |
| SCAFFOLD | - | - | - | - | - | - | - | - | 6 | 0.598 |
| FedCurv | 774 | - | - | 0.316 | 3 | 18 | - | 0.882 | 1 | 0.614 |
| **FedReg** | **248** | **612** | **796** | **0.502** | **2** | **14** | **53** | **0.898** | **1** | **0.673** |

shown in Figure 2, the increases of the loss are significantly lower in FedReg than those in FedAvg, suggesting that although FedAvg and FedReg both forget some of the learned knowledge, the forgetting issue is less severe in FedReg. To further explore the role of the generated pseudo data in the alleviation of forgetting, the values of the empirical Fisher information (Ly et al., 2017) in the pseudo data and previous training data are compared. As shown in Figure 2, the values of the pseudo data and previous training data are significantly correlated. Following the Laplace approximation (MacKay, 1992; Kirkpatrick et al., 2017), if we approximate the posterior distribution of the optimal parameters on a dataset by a normal distribution centered at $\boldsymbol{\theta}^{(t-1)}$ with a diagonal precision matrix, then the precision matrix can be approximated by the empirical Fisher information. The similarity in the empirical Fisher information suggests that a model with high performance on the pseudo data has a relatively high probability to perform well on previous training data. Therefore, regularizing the parameters with the pseudo data can help reduce the performance degradation on previous training data and alleviate the knowledge forgetting issue.

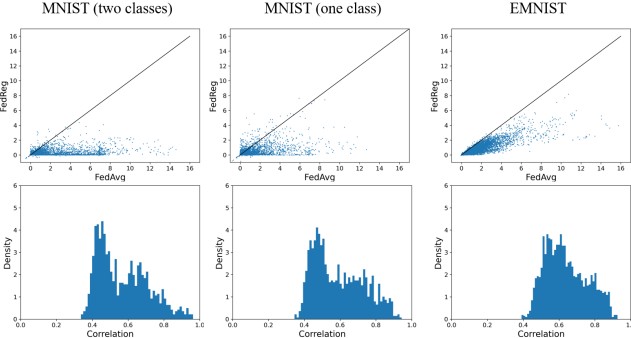

Figure 2: Increase of the loss (top row) concerning previous training data and distribution of the correlation (bottom row) between the empirical Fisher information in the pseudo data and that in the previous training data at the other clients.

## 4.4 THE IMPACT OF STEP SIZE IN GENERATING PSEUDO DATA

The impact of the hyper-parameter $\eta_s$, which determines the distance between the pseudo data and the local data, is explored here. Intuitively, when $\eta_s$ is too large, the distance between the generated pseudo data and the local data will be too large to effectively regularize the model parameters. On the other hand, when $\eta_s$ is too small, the conflicting information from an inaccurate global model will slow down the convergence speed. This intuition is empirically confirmed by the experiments presented in supplementary Figure C.2. On the EMNIST and CIFAR-10 (one class) datasets, when $\eta_s$ is too small, it takes more communication rounds to achieve the same accuracy. As $\eta_s$ increases, the number of communication rounds decreases to a minimum and then bounces back, indicating that a less effective regularization on the model parameters in the local training stage has been reached.

### 4.5 PROTECTION OF PRIVACY

The protection of private information is one of the most important concerns in the FL paradigm, especially in medical applications. To illustrate the privacy security of FedReg with MG, gradient inversion attacks are used to recover information from updated parameters in classification problems. Under a comparable degree of privacy protection, the resultant model performance is compared between FedReg with MG and the baseline methods with DPSGD (Abadi et al., 2016). Using gradient inversion attacks, the quality of images recovered from the updated parameters in a local step is examined. Note that when only one local training step is performed in each round, the updated parameters in FedAvg, FedProx, FedCurv, and SGD are all the same, leading to the same images recovered from these methods. We do not include SCAFFOLD here since its performance was significantly inferior to the other methods in the above experiments. As shown in supplementary Figures C.3 and C.4, for some images sampled from the EMNIST, CIFAR-10 and COVID-19 CT datasets, the quality of the images recovered from FedReg with MG is significantly worse than those recovered from the baseline methods, exhibiting a better privacy protection capability of FedReg with MG. To compare the impact of privacy protection on model performance, we tune the noise level in DPSGD to generate images with similar quality as measured by PSNR (Hore & Ziou, 2010) and compare the performance of the resultant models. The results are shown in Table 4. It can be seen that the protection of private information with DP costs a large performance degradation in the baseline methods, while the performance of FedReg with MG is degraded much less when maintaining a similar privacy protection level. Note that the CT images used in the above experiments contain some privacy-sensitive information, including the date of birth of the patient and the date when the image was taken. In the basic FedAvg, when a 10-layer neural network with 9 layers of CNNs and one layer of FC is used, such information can be easily recovered with gradient inversion attacks. When DP is applied, although such time information could be protected, the resultant model suffers from severe performance degradation, as shown in Table 4. In contrast, FedReg with MG is able to protect the sensitive time information with only mild model performance decay.

Table 4: Performance comparison between the resultant models when DPSGD is applied to the baseline methods and when MG is employed in FedReg. In each case, the final accuracy on the respective test data is reported.

| Method | EMNIST | CIFAR-10 (uniform) | CIFAR-10 (one class) | CT images of COVID-19 |
|---|---|---|---|---|
| DPSGD | 0.932 | 0.285 | 0.233 | 0.533 |
| FedAvg with DPSGD | 0.882 | 0.391 | 0.157 | 0.570 |
| FedProx with DPSGD | 0.881 | 0.392 | 0.156 | 0.570 |
| FedCurv with DPSGD | 0.880 | 0.396 | 0.179 | 0.586 |
| FedReg with MG | **0.986** | **0.703** | **0.599** | **0.657** |

## 5 CONCLUSIONS AND FUTURE WORK

In this work, we proposed a novel algorithm, FedReg, to accelerate the convergence speed of FL by alleviating the catastrophic forgetting issue in the local training stage. Pseudo data are generated to carry knowledge about previous training data learned by the global model without incurring extra communication costs or accessing data provided at the other clients. Our extensive experiments show that the generated pseudo data contain similar Fisher information as the previous training data at the other clients, and thus the forgetting issue can be alleviated by regularizing the locally trained parameters with the pseudo data. The pseudo data can also be used to defend against gradient inversion attacks in classification problems with only mild performance decay on the resultant model compared with DP. Although FedReg exhibits competitive performance in accelerating convergence speeds, its performance is influenced by hyper-parameters. Automatic tuning of the hyper-parameters could make FedReg easier to use. Meanwhile, the question of how to modify gradients in FedReg to effectively protect privacy in regression problems remains to be explored. Moreover, a rigorous proof of the convergence rate of FedReg is of theoretical interest. We leave these questions to future work.

## 6 ACKNOWLEDGEMENT

This work has been supported in part by the US National Institute of Health grant 1R01NS125018, the National Key Research and Development Program of China grant 2018YFC0910404 and the Guoqiang Institute of Tsinghua University with grant no. 2019GQG1.

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

# A  SUPPLEMENTARY METHOD

## A.1  TAYLOR EXPANSION

Expanding both sides of the inequality (4) with $\boldsymbol{\theta}_\beta^{(t,i)} = \frac{\boldsymbol{\theta}^{(t,i)} + \boldsymbol{\theta}^{(t-1)}}{2}$, we obtain

$$
\sum_{\boldsymbol{d}^s \in \mathbb{D}_i^s} L_{\boldsymbol{\theta}^{(t,i)}}\left(\boldsymbol{d}^s\right) = \sum_{\boldsymbol{d}^s \in \mathbb{D}_i^s} L_{\boldsymbol{\theta}_\beta^{(t,i)}}\left(\boldsymbol{d}^s\right) + \left(\boldsymbol{\theta}^{(t,i)} - \boldsymbol{\theta}_\beta^{(t,i)}\right)^T \nabla_{\boldsymbol{\theta}_\beta^{(t,i)}} \sum_{\boldsymbol{d}^s \in \mathbb{D}_i^s} L_{\boldsymbol{\theta}_\beta^{(t,i)}}\left(\boldsymbol{d}^s\right)
$$

$$
+ \frac{1}{2}\left(\boldsymbol{\theta}^{(t,i)} - \boldsymbol{\theta}_\beta^{(t,i)}\right)^T \nabla_{\boldsymbol{\theta}_\beta^{(t,i)}}^2 \left(\sum_{\boldsymbol{d}^s \in \mathbb{D}_i^s} L_{\boldsymbol{\theta}_\beta^{(t,i)}}\left(\boldsymbol{d}^s\right)\right)\left(\boldsymbol{\theta}^{(t,i)} - \boldsymbol{\theta}_\beta^{(t,i)}\right)
$$

$$
+ \mathcal{O}\left(\|\boldsymbol{\theta}_\beta^{(t,i)} - \boldsymbol{\theta}^{(t,i)}\|^3\right),
$$

$$
\sum_{\boldsymbol{d}^s \in \mathbb{D}_i^s} L_{\boldsymbol{\theta}^{(t-1)}}\left(\boldsymbol{d}^s\right) = \sum_{\boldsymbol{d}^s \in \mathbb{D}_i^s} L_{\boldsymbol{\theta}_\beta^{(t,i)}}\left(\boldsymbol{d}^s\right) + \left(\boldsymbol{\theta}^{(t-1)} - \boldsymbol{\theta}_\beta^{(t,i)}\right)^T \nabla_{\boldsymbol{\theta}_\beta^{(t,i)}} \sum_{\boldsymbol{d}^s \in \mathbb{D}_i^s} L_{\boldsymbol{\theta}_\beta^{(t,i)}}\left(\boldsymbol{d}^s\right)
$$

$$
+ \frac{1}{2}\left(\boldsymbol{\theta}^{(t-1)} - \boldsymbol{\theta}_\beta^{(t,i)}\right)^T \nabla_{\boldsymbol{\theta}_\beta^{(t,i)}}^2 \left(\sum_{\boldsymbol{d}^s \in \mathbb{D}_i^s} L_{\boldsymbol{\theta}_\beta^{(t,i)}}(\boldsymbol{d}^s)\right)\left(\boldsymbol{\theta}^{(t-1)} - \boldsymbol{\theta}_\beta^{(t,i)}\right)
$$

$$
+ \mathcal{O}\left(\|\boldsymbol{\theta}_\beta^{(t,i)} - \boldsymbol{\theta}^{(t-1)}\|^3\right).
$$

Substituting both sides of (4), dividing them by $|\mathbb{D}_i^s|$ and ignoring all high order terms, then only the first order terms are left and the thus inequality (6) is obtained, *i.e.*,

$$
\left(\boldsymbol{\theta}^{(t-1)} - \boldsymbol{\theta}^{(t,i)}\right)^T \nabla_{\boldsymbol{\theta}_\beta^{(t,i)}} \frac{1}{|\mathbb{D}_i^s|} \sum_{\boldsymbol{d}^s \in \mathbb{D}_i^s} L_{\boldsymbol{\theta}_\beta^{(t,i)}}(\boldsymbol{d}^s) \geq 0.
$$

The inequality (7) can be derived from (5) in a similar way.

## A.2  SOLVING THE CONSTRAINT OPTIMIZATION

The constraint optimization problems in (8) and (9) can be solved by the Lagrange multiplier method. More specifically, for the problem (8), *i.e.*,

$$
\boldsymbol{\theta}^{(t,i)'} = \arg\min_{\boldsymbol{\theta}} \|\boldsymbol{\theta} - \boldsymbol{\theta}^{(t,i)}\|^2
$$

$$
s.t. \left(\boldsymbol{\theta}^{(t-1)} - \boldsymbol{\theta}\right)^T \boldsymbol{g}_\beta(\mathbb{D}_i^s) \geq 0,
$$

its Lagrangian can be written as

$$
\mathcal{L}(\boldsymbol{\theta}, \alpha) = \|\boldsymbol{\theta} - \boldsymbol{\theta}^{(t,i)}\|^2 + \alpha\left(\boldsymbol{\theta} - \boldsymbol{\theta}^{(t-1)}\right)^T \boldsymbol{g}_\beta\left(\mathbb{D}_i^s\right),
$$

where $\alpha \geq 0$. The dual problem is

$$
\mathcal{D}(\alpha) = \min_{\boldsymbol{\theta}} \mathcal{L}(\boldsymbol{\theta}, \alpha).
$$

Setting the derivatives of $\mathcal{L}(\boldsymbol{\theta}, \alpha)$ w.r.t. $\boldsymbol{\theta}$ to zero, the value of $\boldsymbol{\theta}^{(t,i)'}$ minimizing $\mathcal{L}(\boldsymbol{\theta}, \alpha)$ is obtained as

$$
2\left(\boldsymbol{\theta}^{(t,i)'} - \boldsymbol{\theta}^{(t,i)}\right) + \alpha\boldsymbol{g}_\beta\left(\mathbb{D}_i^s\right) = 0,
$$

$$
\boldsymbol{\theta}^{(t,i)'} = \boldsymbol{\theta}^{(t,i)} - \tfrac{1}{2}\alpha\boldsymbol{g}_\beta\left(\mathbb{D}_i^s\right).
$$

Thus, $\mathcal{D}(\alpha)$ can be simplified as

$$
\mathcal{D}(\alpha) = -\frac{1}{4}\alpha^2 \boldsymbol{g}_\beta(\mathbb{D}_i^s)^T \boldsymbol{g}_\beta(\mathbb{D}_i^s) + \alpha(\boldsymbol{\theta}^{(t,i)} - \boldsymbol{\theta}^{(t-1)})^T \boldsymbol{g}_\beta\left(\mathbb{D}_i^s\right).
$$

Solving $\alpha^* = \arg\max \mathcal{D}(\alpha)$, the value of $\alpha^*$ is

$$
\alpha^* = \max\left(\frac{2\left(\boldsymbol{\theta}^{(t,i)} - \boldsymbol{\theta}^{(t-1)}\right)^T \boldsymbol{g}_\beta\left(\mathbb{D}_i^s\right)}{\boldsymbol{g}_\beta\left(\mathbb{D}_i^s\right)^T \boldsymbol{g}_\beta\left(\mathbb{D}_i^s\right)}, 0\right).
$$

Thus, the value of $\boldsymbol{\theta}^{(t,i)'}$ is

$$
\boldsymbol{\theta}^{(t,i)'} = \boldsymbol{\theta}^{(t,i)} - \max\left(\frac{\left(\boldsymbol{\theta}^{(t,i)} - \boldsymbol{\theta}^{(t-1)}\right)^T \boldsymbol{g}_\beta\left(\mathbb{D}_i^s\right)}{\boldsymbol{g}_\beta\left(\mathbb{D}_i^s\right)^T \boldsymbol{g}_\beta\left(\mathbb{D}_i^s\right)}, 0\right) \boldsymbol{g}_\beta\left(\mathbb{D}_i^s\right).
$$

The constraint optimization problem 9 can be solved similarly.

## B    Experimental details

In all the experiments, the learning rates for different methods are optimized by grid search, the optimal weight for the proximal term in FedProx is searched among $\{1.0, 0.1, 0.01, 0.001\}$, the optimal $\lambda$ for FedCurv is searched among $\{0.001, 0.0001, 0.00001\}$, and the hyper-parameters ($\gamma$ and $\eta_s$) in FedReg are optimized by grid search as well. The number of epochs in the local training stage is optimized in FedAvg and applied to the other methods.

### B.1    Computation of $loss^{(t-1)}$ and $loss^{(t)}$

Formally, the value of $loss^{(t-1)}$ for a client $i \in \mathbb{C}^{(t-1)}$ is computed as

$$loss^{(t-1)}(i) = \frac{1}{|\mathbb{D}_i|} \sum_{\boldsymbol{d} \in \mathbb{D}_i} L_{\boldsymbol{\theta}^{(t-1)}}(\boldsymbol{d}),$$

and the value of $loss^{(t)}$ is computed as

$$loss^{(t)}(i) = \frac{1}{|\mathbb{D}_i|} \sum_{\boldsymbol{d} \in \mathbb{D}_i} \frac{\sum_{j \in \mathbb{C}^{(t)}}(L_{\boldsymbol{\theta}^{(t,j)}}(\boldsymbol{d}))}{|\mathbb{C}^{(t)}|},$$

where $\boldsymbol{\theta}^{(t,j)}$ is the parameters sent from client $j$ in round $t$.

### B.2    Comparison of convergence rates

The hyper-parameters in the experiments are listed below:

- **MNIST.** Ten clients are sampled in each training round and 500 rounds are run in total with a batch size of 10 so that all training data are utilized once on average. The learning rate in all methods is 0.1. On MNIST (two classes), in the local training stage, the local data are processed in 40 epochs. The weight for the proximal term in FedProx is 0.001 and $\lambda$ in FedCurv is $10^{-4}$. In FedReg, $\gamma = 0.4$ and $\eta_s = 0.2$. On MNIST (one classes), the local data are processed in 20 epochs. The weight for the proximal term in FedProx is 0.01 and $\lambda$ in FedCurv is $10^{-4}$. In FedReg, $\gamma = 0.3$ and $\eta_s = 0.2$.

- **EMNIST.** 20 clients are sampled in each training round, and in the local training stage, the local data are processed in 20 epochs with a batch size of 24. 500 rounds are run in total so that all the training data are utilized once on average. The learning rate in SCAFFOLD is 0.1 and in other methods, it is 0.2. The weight for the proximal term in FedProx is 0.001 and $\lambda$ in FedCurv is $10^{-4}$. In FedReg, $\gamma = 0.4$ and $\eta_s = 0.05$.

- **CIFAR-10.** Following the experimental settings in (Rothchild et al., 2020), 240 rounds are run in total with 100 clients sampled in each training round, and in the local training stage, the batch size is 5. On CIFAR-10 (uniform), the local data are processed in 30 epochs. The learning rate is 0.05 in FedAvg, FedCurv and FedProx, 0.1 in SGD and FedReg, and 0.01 in SCAFFOLD respectively. The weight for the proximal term in FedProx is 0.001 and $\lambda$ in FedCurv is $10^{-5}$. In FedReg, $\gamma = 0.5$ and $\eta_s = 0.1$. On CIFAR-10 (one class), the local data are processed in 20 epochs. The learning rate is 0.05 in FedAvg, FedCurv, FedProx, SCAFFOLD and FedReg, and 0.1 in SGD, respectively. The weight for the proximal term in FedProx is 0.01 and $\lambda$ in FedCurv is $10^{-3}$. In FedReg, $\gamma = 0.5$ and $\eta_s = 0.1$.

- **CIFAR-100.** When ResNet-9 is adopted, 1,200 rounds are run in total with 100 clients sampled in each training round, and in the local training stage, the local data are processed in 10 epochs with a batch size of 1. The learning rate is 0.05 in FedAvg, FedCurv and FedProx, and 0.1 in SGD and FedReg, respectively. The weight for the proximal term in FedProx is 0.01 and $\lambda = 10^{-3}$ in FedCurv. In FedReg, $\gamma = 0.25$ and $\eta_s = 0.1$. When the pre-trained transformer is adopted, 100 rounds are run in total with 500 clients sampled in each training round, and in the local training stage, the local data are processed in 5 epochs with a batch size of 1. The learning rate is 0.1 in all methods. The weight for the proximal term in FedProx is 0.001 and $\lambda = 10^{-5}$ in FedCurv. In FedReg, $\gamma = 0.02$ and $\eta_s = 10^{-2}$. Note that as SCAFFOLD is too memory-consuming to run on CIFAR-100 given our computational resources, the results of SCAFFOLD on CIFAR-100 are not reported.

- **CT images related to COVID-19.** In our experiments, ten rounds are run in total and in each round, 10 clients are sampled. In the local training stage, the local data are processed in 20 epochs with a batch-size of 10. The learning rate is $1 \times 10^{-3}$ in FedAvg, FedCurv and FedProx, $5 \times 10^{-4}$ in SCAFFOLD, $1 \times 10^{-4}$ in SGD, and $5 \times 10^{-3}$ in FedReg, respectively. The weight for the proximal term in FedProx is $1.0$ and $\lambda$ in FedCurv is $10^{-4}$. In FedReg, $\gamma = 0.5$ and $\eta_s = 10^{-6}$. Note that group-normalization (Wu & He, 2018) is used in the DenseNet-121 network as the batch size is small in the local training stage.

- **Landmarks-User-160k.** The training data in this dataset has 164,172 images of 2,028 landmarks from 1,262 users, and the test data contains 19,526 images. Each user has images of 29 classes on average. A DenseNet-121 network is trained on this dataset. 2,000 rounds are run in total, and in each round 100 clients are sampled. In the local training stage, the local data is processed in 40 epochs. The learning rate is set as $0.1$ in SGD, FedAvg, FedReg, and pFedGP (Achituve et al., 2021). As pFedGP is designed for personalized federated learning scenarios and trains an independent GP-tree for each client, to apply it to this dataset, a small global dataset is built by sampling 2,028 images from all training data, which covers all the classes contained in Landmarks-User-160k. In each round, the deep kernel, which is the DenseNet-121 network without the classification layer, is trained on each client and aggregated in the server. Then, a global GP-tree is trained on the sampled global sub-dataset to classify the images in the test data.

### B.3 ALLEVIATION OF CATASTROPHIC FORGETTING

The hyper-parameters used here, except $\gamma$ in FedReg, are the same as the values used in the previous section.

- **Computation of loss increment.** To fairly compare the forgetting issue in FedReg and FedAvg, $\gamma$ is set to be $1.0$ in FedReg, and in each round, the client parameters are initialized with the parameters obtained in FedReg to avoid a large discrepancy between the values of $loss^{(t-1)}$ computed in FedReg and FedAvg. For a client $i \in \mathbb{C}^{(t-1)}$, the loss increase is computed as $loss^{(t)}(i) - loss^{(t-1)}(i)$.

- **The computation of Fisher information.** Following (Kunstner et al., 2019), in round $t$ of client $i \in \mathbb{C}^{(t)}$, for the previous training data $\mathbb{D}^{(t-)} = \cup_{j \in \mathbb{C}^{(t-)}} D_j$, where $\mathbb{C}^{(t-)} = \cup_{s \in [t-10, t-1]} \mathbb{C}^{(s)}$, the empirical Fisher information is computed as

$$\text{fisher}_{\boldsymbol{\theta}^{(t,i)}}(\mathbb{D}^{(t-)}) = \frac{1}{|\mathbb{D}^{(t-)}|} \sum_{(\boldsymbol{x}, \boldsymbol{y}) \in \mathbb{D}^{(t-)}} \left( \nabla_{\boldsymbol{\theta}^{(t,i)}} \boldsymbol{y}^T \log \boldsymbol{p}_{\boldsymbol{\theta}^{(t,i)}} \left( \boldsymbol{y} | \boldsymbol{x} \right) \right)^2,$$

Similarly for the pseudo data $\mathbb{D}_i^s$, the empirical Fisher information is computed as

$$\text{fisher}_{\boldsymbol{\theta}^{(t,i)}}(\mathbb{D}_i^s) = \frac{1}{|\mathbb{D}_i^s|} \sum_{(\boldsymbol{x}^s, \boldsymbol{y}^s) \in \mathbb{D}_i^s} \left( \nabla_{\boldsymbol{\theta}^{(t,i)}} \boldsymbol{y}^{sT} \log \boldsymbol{p}_{\boldsymbol{\theta}^{(t,i)}} \left( \boldsymbol{y}^s | \boldsymbol{x}^s \right) \right)^2.$$

Note that considering the generation process of the pseudo data, the parameter values $\boldsymbol{\theta}^{(t,i)}$ used to compute Fisher information are the same local model parameters after a local training step.

### B.4 PROTECTION OF PRIVATE INFORMATION

The target of gradient inversion attacks in (Geiping et al., 2020) and (Zhu et al., 2019) is to find

$$\boldsymbol{d}^* = \arg \min_{\boldsymbol{d}} \text{Distance} \left( \Delta \boldsymbol{\theta} \left( \boldsymbol{\theta}^{(t)}, \boldsymbol{d}_{true} \right), \Delta \boldsymbol{\theta} \left( \boldsymbol{\theta}^{(t)}, \boldsymbol{d} \right) \right) + w \text{TV} \left( \boldsymbol{d} \right)$$

where $\Delta \boldsymbol{\theta} \left( \boldsymbol{\theta}^{(t)}, \cdot \right)$ denotes the amounts (updates) that the parameters have changed from $\boldsymbol{\theta}^{(t)}$ with the corresponding data, $\text{TV} \left( \boldsymbol{d} \right)$ the total variation of $\boldsymbol{d}$, and $\boldsymbol{d}_{true}$ a data point on the client. The distance function could be cosine distance (Geiping et al., 2020) or $L2$-distance (Zhu et al., 2019). Note that the method proposed in (Zhu et al., 2019) sets $w = 0$. To consider the worst case, in our attacking experiments, the parameters are only trained for one step with each data point, in

which case the values of $\Delta\boldsymbol{\theta}\left(\boldsymbol{\theta}^{(t)}, \boldsymbol{d}\right)$ obtained in FedAvg, FedCurv, FedProx and SGD are exactly the same. 32,000 attacking iterations are used in all experiments and both distance functions (*i.e.*, cosine and $L2$) are considered. Adam (Kingma & Ba, 2014) optimizer is used, which performs better than L-BFGS empirically. The learning rate and the weight for the total variation are optimized in (the basic) FedAvg and applied to the other methods. The learning rate decaying scheme follows the method provided in (Geiping et al., 2020).

More details concerning each dataset are discussed below.

- **EMNIST.** The images to be recovered are randomly sampled. The model architecture and hyper-parameters used in FedAvg, FedProx, FedCurv, SGD, and FedReg are the same as the ones used in section B.2. In the baseline methods with DPSGD, $C = 1.0$ and $\sigma = 0.01$, and other hyper-parameters are the same as the ones used in the corresponding baseline methods (without DPSGD). In FedReg with MG, the learning rate and the value of $\gamma$ are the same as the ones used in FedReg, and $\eta_s = 0.03$. In the attacking algorithm, the initial learning rate is $1.0$, the distance function is $L2$-distance, and the weight for the total variation term is $10^{-8}$.

- **CIFAR-10.** The images to be recovered are randomly sampled. The model architecture and hyper-parameters used in FedAvg, FedProx, FedCurv, SGD, and FedReg are the same as the ones used in section B.2. In the baseline methods with DPSGD, $C = 1.0$ and $\sigma = 0.05$, and the other hyper-parameters are the same as the ones used in the corresponding baseline methods (without DPSGD). In FedReg with MG, the learning rate and the value of $\gamma$ are the same as the one used in FedReg, and $\eta_s = 0.001$. In the attacking algorithm, the initial learning rate is $1.0$, the distance function is cosine distance, and the weight for the total variation term is $10^{-6}$.

- **CT images.** Only the images containing time information are used here. As DenseNet-121 is too complex to be attacked, we use a ten-layer neural network with 9 layers of CNNs and one layer of FC instead. 10 rounds are run in total and in each round, 10 clients are sampled. In the local training stage, the local data are processed in 20 epochs. The learning rate is $5 \times 10^{-4}$ in SGD, FedAvg, FedProx and FedReg, and $1 \times 10^{-3}$ in FedCurv, respectively. The weight for the proximal term in FedProx is $0.1$ and $\lambda$ in FedCurv is $0.001$. In FedReg, $\gamma = 0.5$ and $\eta_s = 10^{-6}$. In the baseline methods with DPSGD, $C = 1.0$ and $\sigma = 0.001$, and other hyper-parameters are the same as the ones used in the corresponding methods (without DPSGD). In FedReg with MG, the learning rate and the values of $\gamma$ and $\eta_s$ are the same as those in FedReg. In the attacking algorithm, the initial learning rate is $1.0$, the distance function is $L2$-distance, and the weight for the total variation term is $10^{-8}$.

## C SUPPLEMENTARY RESULTS

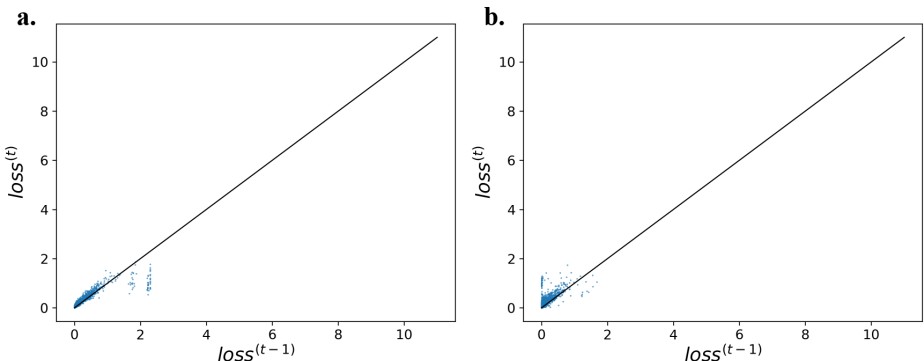

Figure C.1: The values of $loss^{(t)}$ and $loss^{(t-1)}$ on homogeneously distributed (a) EMNIST and (b) MNIST, where the whole dataset is uniformly partitioned into $10,000$ clients in EMNIST and $2,000$ clients in MNIST, respectively. It can be seen that the values of $loss^{(t)}$ are comparable to those of $loss^{(t-1)}$, indicating not much previous training data has been forgotten.

Table C.1: Comparison results on Landmarks-User-160k after 2,000 rounds.

| Method | Landmarks-User-160k | | | |
| | $R_{0.5}$ | $R_{0.9}$ | $R_{1.0}$ | ACC |
|---|---|---|---|---|
| SGD | 1178 | 1824 | 1953 | 0.045 |
| FedAvg | 84 | 169 | 190 | 0.145 |
| pFedGP | - | - | - | 0.005 |
| FedReg | **50** | **101** | **127** | **0.160** |

Table C.2: Average wall-clock time for running a local training stage on MNIST and EMNIST. The GPUs are 1080 Ti.

| Method | MNIST (two classes) | EMNIST |
|---|---|---|
| FedAvg | 0.7 s | 0.3 s |
| FedProx | 0.7 s | 0.3 s |
| FedCurv | 0.7 s | 0.3 s |
| FedReg | 1.1 s | 0.5 s |

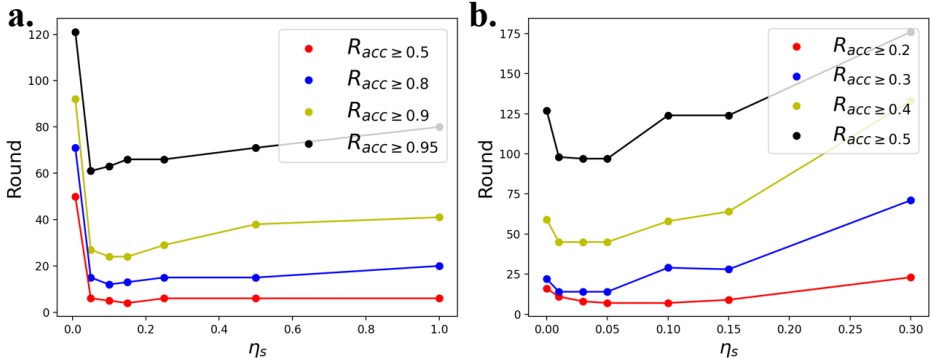

Figure C.2: Impact of $\eta_s$ on the model performance on EMNIST (a) and CIFAR-10 (one class) (b). The curve $R_{acc \geq a}$ shows the minimum number of rounds required to reach the accuracy of $a$.

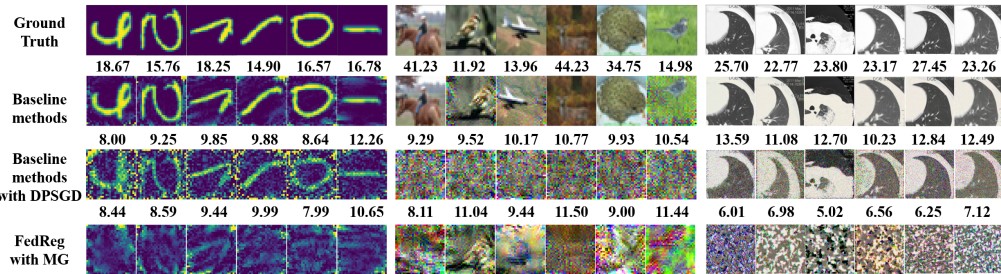

Figure C.3: Images recovered from updated parameters computed on the EMNIST, CIFAR-10, and COVID-19 CT datasets, and their corresponding PSNR scores. Note that in the attacking process, when only one local training step is performed in each round, the images recovered from the baseline methods FedAvg, FedProx, FedCurv, and SGD, are all the same. High-resolution versions of the CT images are shown in Figure C.4.

## D  ALGORITHM

---

**Algorithm D.1** Training Procedure of FedReg.

---

**Input**: $K, T, \{\mathbb{D}_i\}_{i \in \mathbb{C}}, \boldsymbol{\theta}^{(0)}, \gamma, \eta_{\boldsymbol{\theta}}, \eta_s, \eta_p, S, E$
**Output**: $\{\boldsymbol{\theta}^{(t)}\}_{t=1}^{T}$
 1: **for** $t = 1, 2, ..., T$ **do**
 2:     Randomly sample $K$ clients $\mathbb{C}^{(t)}$
 3:     **for** $i \in \mathbb{C}^{(t)}$ **do**
 4:         $\boldsymbol{\theta}^{(t,i)} \leftarrow \boldsymbol{\theta}^{(t-1)}$, generate $\mathbb{D}_i^s$ and $\mathbb{D}_i^p$ with $\eta_s, \eta_p$ and $E$
 5:         **for** $s = 1, 2, ..., S$ **do**
 6:             Compute $\boldsymbol{g}_\gamma(\mathbb{D}_i), \boldsymbol{\theta}^{(t,i)} \leftarrow \boldsymbol{\theta}^{(t,i)} - \eta_{\boldsymbol{\theta}} \boldsymbol{g}_\gamma(\mathbb{D}_i), \boldsymbol{\theta}_\beta^{(t,i)} \leftarrow 0.5\left(\boldsymbol{\theta}^{(t,i)} + \boldsymbol{\theta}^{(t-1)}\right)$
 7:             Compute $\boldsymbol{g}_\beta(\mathbb{D}_i^s), \boldsymbol{g}_\beta(\mathbb{D}_i^p), w_s$ and $w_p, \boldsymbol{\theta}^{(t,i)} \leftarrow \boldsymbol{\theta}^{(t,i)} - w_s \boldsymbol{g}_\beta(\mathbb{D}_i^s) - w_p \boldsymbol{g}_\beta(\mathbb{D}_i^p)$
 8:         **end for**
 9:         Send $\boldsymbol{\theta}^{(t,i)}$ to the server
10:     **end for**
11:     $\boldsymbol{\theta}^{(t)} \leftarrow \frac{1}{K} \sum_{i \in \mathbb{C}^{(t)}} \boldsymbol{\theta}^{(t,i)}$
12: **end for**
13: **return** $\{\boldsymbol{\theta}^{(t)}\}_{t=1}^{T}$

---

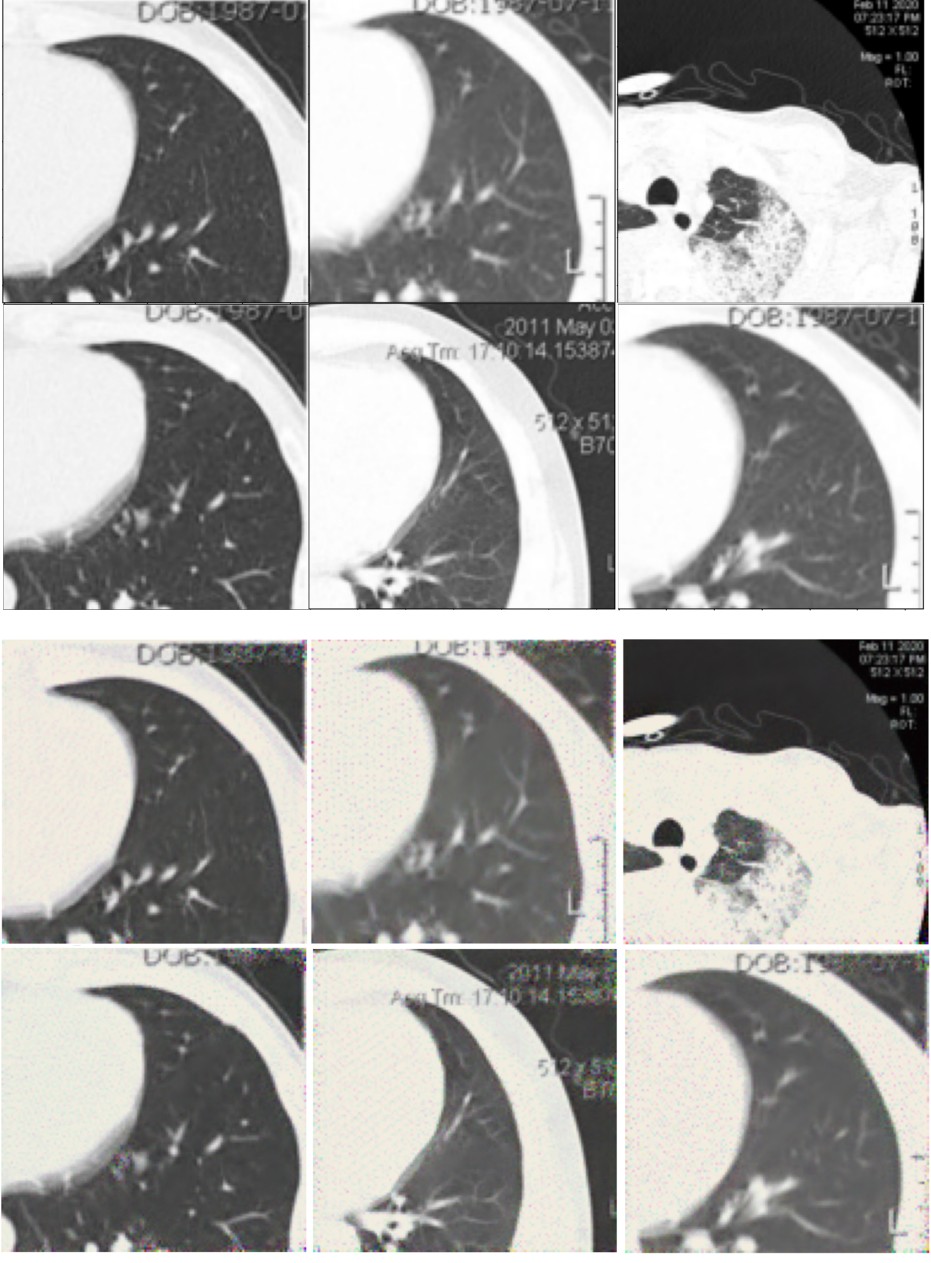

Figure C.4: High-resolution versions of the CT images (rows 1 and 2) and the images recovered from FedAvg (rows 3 and 4). Note that the time stamps located at the upper right corners of the recovered images are visible.

