# OpenReview forum: "Acceleration of Federated Learning with Alleviated Forgetting in Local Training"
_ICLR.cc/2022/Conference — ICLR 2022 Poster_

### Official Review · Reviewer_YGd2 · 2021-10-29

**Correctness:** 3
**Technical Novelty And Significance:** 3
**Empirical Novelty And Significance:** 3
**Recommendation:** 6
**Confidence:** 4

**Main Review:**

Strengths:
- The problem presented in the paper is well-motivated and the proposed solution for handling it looks promising.
- Generating the pseudo-datasets at the client using the global model is a good idea and avoids potential private information sharing.
- The paper is technically correct. I have gone through its derivations and did not find any flaws.
- The results of the method are strong compared to the baseline methods examined.
- The protection of privacy is a nice addition to the paper which distinguishes it from standard papers in the field.
- For the most part, the paper is written clearly.
- Full experimental details and code were provided.

Weaknesses:
- Some parts of the method, in my opinion, could be motivated and explained better. For example:
     1. Why did you choose to generate the pseudo-datasets via adversarial training? Several alternatives for generating and using pseudo data were proposed in the literature (e.g., [1, 2, 3, 4]). I think the authors should address this line of research in the paper.
     2. Why did you need to perform Taylor expansion to Eq. 4 & 5? Why can't they be used directly?
     3. Given the approximations made, can you please clarify what is the effect of replacing the inequality sign in Eq. 4  with equality (two lines below Eq. 10)?

- Having a shared model only is fine, but the paper misses an important line of work of personalized federated learning which combats the data heterogeneity using personalized components of the model. I believe that the issue of catastrophic forgetting is less severe in these models as the personalized components should make up for that. Did you check that? I think that a comparison to recent strong methods (such as [1]) is missing.

General Questions:
- Regarding the training procedure, why did you choose to perform one step on $\mathbb{D}_i$ followed by one step on $\mathbb{D}_i^s$ and $\mathbb{D}_i^p$?

- Some experimental details are not clear. For example, how did you choose the hyper-parameters based on the grid search? Did you have a validation set?


[1] Achituve, I., Shamsian, A., Navon, A., Chechik, G., & Fetaya, E. (2021). Personalized Federated Learning with Gaussian Processes. arXiv preprint arXiv:2106.15482.
[2] Luo, M., Chen, F., Hu, D., Zhang, Y., Liang, J., & Feng, J. (2021). No Fear of Heterogeneity: Classifier Calibration for Federated Learning with Non-IID Data. arXiv preprint arXiv:2106.05001.
[3] Hao, W., El-Khamy, M., Lee, J., Zhang, J., Liang, K. J., Chen, C., & Duke, L. C. (2021). Towards Fair Federated Learning with Zero-Shot Data Augmentation. In Proceedings of the IEEE/CVF Conference on Computer Vision and Pattern Recognition (pp. 3310-3319).
[4] Goetz, J., & Tewari, A. (2020). Federated learning via synthetic data. arXiv preprint arXiv:2008.04489.

**Summary Of The Paper:**

The authors claim that in non-iid federated learning setups models experience catastrophic forgetting of previously encountered data. As a result, the performance of the model degrades and the convergence is slower. As a solution, the authors propose FedReg, a method to regularize the model with pseudo-data generated from the local data of the client. Two datasets are generated using the fast gradient sign method. The first dataset, termed pseudo-data, aims at simulating data of other clients in the system. The second dataset, termed perturbed data, aims at retaining the model power on the local dataset of the current client. These datasets are used along with the original data of the client to update the local model parameters. The authors further introduce a method to modify the gradients to enhance the protection of privacy. The authors compared their method against several baselines on MNIST, EMNIST, CIFAR-10, CIFAR-100, and CT images. The comparisons show that FedReg achieves significant improvements in accuracy and convergence, reduction in catastrophic forgetting, and better protection of privacy compared to baseline methods.

**Summary Of The Review:**

Overall I think that this paper could be a nice contribution. I think that the experimental section can be improved, and some important references are missing. Upon addressing my concerns I am willing to reevaluate the paper.

---

> ### Author Response · Authors · 2021-11-23
> **Response to Reviewer YGd2**
>
> Thanks so much for your careful review and plentiful encouraging comments. We respond in detail to each of your comments concerning the weakness of our manuscript and general questions below. For your convenience, we have also highlighted the main changes in the revised manuscript.
>
> 1.  Regarding point 1, the motivation behind the generation of pseudo data is mentioned in the (slightly expanded) discussion immediately following Assumption 2 and explained in more detail in Section 3.1. Briefly, we hope that the generated pseudo data are close to the local data while having very different labels, and adversarial examples happen to satisfy such a requirement. Although the alternative methods suggest by the reviewer might be viable, our current strategy is preferred given our specific aim of reducing the communication costs of FL by alleviating the forgetting issue in the local training stage. Since our submission has already reached the page limit of ICLR, we will try to add more discussions of alternative methods for generating pseudo data in the final version.
>
>     Regarding point 2, equations 4 and 5 are almost impossible to solve directly given the complexity of deep learning models. Therefore, Taylor expansion is used to simplify the computation.
>
>     Regarding point 3, what we mean is that although inequality 4 can in fact be strengthened to an equality theoretically, our method still uses inequality because it would make the training process more robust and result in non-negative values of $w_s$ and $w_p$. We have revised the discussion below equation 10 slightly to make this clearer.
>
> 2. Thanks for the comment and bringing personalized FL methods to our attention. Since we aim at computing an optimal model for the global data, we have not considered the personalized methods for two reasons. First, since it is easy to personalize a well-trained global model by many techniques such as fine-tuning, we thought that training a good global model is perhaps more essential in FL. Second, including the personalized methods in our experiments would perhaps be unfair for these methods, since the test data follows the distribution of the global data instead of the local data of any individual client. Nevertheless, supplementary Table C.1 of our revised manuscript includes a comparison with pFedGP [1] on a real-world dataset, Landmarks-User-160k [2, 3], and it shows that pFedGP performed very poorly. This could be due to the fact that the local data on each client only has images from 29 of the 2,028 classes on average and a personalized model is unable to provide effective updates to the shared deep learning kernel.
>
> 3. We update the parameters in such a sequence because compared with performing only one step on $D_i^s$ and $D_i^p$ at the end of the local training, iteratively performing a step on $D_i$ followed by a step on $D_i^s$ and $D_i^p$ is more conducive for the model to find a balance between fitting the local data and satisfying the regularization requirements.
>
> 4. Validation sets are not used in our experiments. We tuned the hyper-parameters by grid search (as mentioned in Appendix B) to find the ones that allow the models to perform the best on each test data.
>
> [1] Achituve, I., Shamsian, A., Navon, A., Chechik, G., & Fetaya, E. (2021). Personalized Federated Learning With Gaussian Processes. Advances in Neural Information Processing Systems, 34.
>
> [2] Lai, F., Dai, Y., Zhu, X., Madhyastha, H. V., & Chowdhury, M. (2021, October). FedScale: Benchmarking model and system performance of federated learning. In Proceedings of the First Workshop on Systems Challenges in Reliable and Secure Federated Learning (pp. 1-3).
>
> [3] Hsu, T. M. H., Qi, H., & Brown, M. (2020). Federated visual classification with real-world data distribution. In Computer Vision–ECCV 2020: 16th European Conference, Glasgow, UK, August 23–28, 2020, Proceedings, Part X 16 (pp. 76-92). Springer International Publishing.

---

> > ### Comment · Reviewer_YGd2 · 2021-11-29
> > **Response to authors' comment**
> >
> > Thank you for the comment. I think that most of the concerns raised by me and other reviewers were fixed. I believe that the paper now, after the revisions, is more clear.
> >
> > Two things that still trouble me. First, the rather limited effort in the hyper-parameter tuning of baseline methods (e.g., different learning rates and a number of epochs can drastically change the performance of methods), and second, more importantly, the use of the test set to optimize the hyper-parameters. Regarding the second issue, it is a bad practice that should be avoided. Furthermore, the results may circulate in the literature which will create a new baseline that is hard to beat and may encourage other papers to follow the same procedure.
> >
> > After careful considerations, I decided to raise the score to 6 since the comparisons and the method in this paper look promising. I believe that the authors should state explicitly in the paper that hyperparameter tuning was done based on the test set. A better approach would be to redo all the experiments and report numbers that were obtained using a valid approach (e.g., validation-based early stopping).

---

> > > ### Author Response · Authors · 2021-12-02
> > > **Further Response:**
> > >
> > > Thanks so much for your positive feedback on our response and constructive suggestions. Given the time constraint, we respond to your second (more important) concern below. We used the test data to tune the hyper-parameters in our work because (1) the experimental data from the literature that were considered here consist of only training and test sets and (2) some of the baseline methods (such as FedProx) tuned their parameters using the test data while the others did not explicitly state how this was done. However, we agree that a validation-based tuning method would be more appropriate, and repeated the experiments in Table 2 using newly tuned parameters. More precisely, for each dataset, we randomly sampled 10% of its training set as its validation set and tuned the hyper-parameters of each method on the dataset by grid search and validation-based early stopping. The new experimental results are summarized in the following table and the details of the resultant hyper-parameters are given below. Clearly, although the performance numbers of each method have fluctuated a bit, the trend remains the same as in Table 2. We plan to redo all the experiments for the final version as much as time permits, even though we believe that all of our conclusions in the manuscript will still stand.
> > >
> > >
> > > |                    | CIFAR-10 (uniform) |                    |                    |                    | CIFAR-10 (one class) |                   |                    |                    |
> > > |:-------------------:|-------------------:|-------------------:|-------------------:|-------------------:|---------------------:|-------------------:|-------------------:|-------------------:|
> > > |                    | $R_{0.5}$| $R_{0.9}$ | ${R_{1.0}}$ | ACC       | $R_{0.5}$| $R_{0.9}$ | $R_{1.0}$ | ACC       |
> > > | SGD                | 40 | 160 | 185 | 0.483 | 39 | 168 | 197 | 0.449 |
> > > | FedAvg             | 4 | 52 | 94 | 0.576 | 108 | -         | -         | 0.371 |
> > > | FedProx            | 6 | 53 | 94 | 0.576 | 108 | -         | -         | 0.373 |
> > > | SCAFFOLD           | 7 | 116 | -         | 0.447 | 88 | -         | -         | 0.23 |
> > > | FedCurv            | 4 | 53 | 94 | 0.569 | 108 | -         | -         | 0.368 |
> > > | FedReg             | **2**| **31**    | **40**    | **0.715**            | **9**     | **58**    | **84**    | **0.616** |
> > >
> > > **Parameter details**.
> > > * On CIFAR-10 (uniform), the local data are processed in 30 epochs with a batch size of 5. The learning rate is $0.05$ in FedAvg, FedCurv and FedProx, $0.1$ in SGD and FedReg, and $0.01$ in SCAFFOLD, respectively. The weight for the proximal term in FedProx is $0.001$ and the value of $\lambda$ in FedCurv is $10^{-5}$. In FedReg, $\gamma=0.5$ and $\eta_s=0.1$.
> > > * On CIFAR-10 (one class), the local data are processed in 20 epochs. The learning rate is $0.05$ in FedAvg, FedCurv, FedProx, SCAFFOLD and FedReg, and $0.1$ in SGD, respectively. The weight for the proximal term in FedProx is $0.01$ and the value of $\lambda$ in FedCurv is $10^{-3}$. In FedReg, $\gamma=0.5$ and $\eta_s=0.1$.

---

### Official Review · Reviewer_9rQY · 2021-10-31

**Correctness:** 2
**Technical Novelty And Significance:** 3
**Empirical Novelty And Significance:** 3
**Recommendation:** 6
**Confidence:** 4

**Main Review:**

Strength:
-Empirical evaluations show strong results, in several experiments
- Idea is novel

Weaknesses:
- My main issue is with the writing of this paper. They claim the main issue is catastrophic forgetting but they don't really show it. They show fig. 1 but should compare to homogenous data where the forgetting shouldn't happen
- Also, they show good results also on homogeneous data (CIFAR-10 uniform) which again points at something else as the reason why their method works
- It is also unclear why the synthetic dataset helps with catastrophic forgetting. Should be explain better
- Assumption 2 is unclear and also I am not sure where exactly do they use it
- Just below eq. 10 they claim that L_{\theta^{t-1}} is optimal but that doesn't seem possible considering eq. 4 (in convex case we would have a fixed point right away).


**Summary Of The Paper:**

The paper addresses the problem of federated learning, they claim the main issue is catastrophic forgetting, which they solve by generating a synthetic dataset per client.

**Summary Of The Review:**

The algorithm seems to work well, but I am skeptical about the alleged problem it solves and that it really addresses it. Writing is the main issue.

---

> ### Author Response · Authors · 2021-11-23
> **Response to Reviewer 9rQY**
>
> Thanks so much for reviewing our submission and providing many insightful comments. We respond to each of your comments concerning the weakness of our manuscript in detail below. For your convenience, we have also highlighted the main changes in the revised manuscript.
>
> 1. Thanks for the suggestion. The values of $Loss^{(t)}$ and $Loss^{(t-1)}$ on homogeneously distributed MNIST and EMNIST have been added as supplementary Figure C.1 of the revised manuscript. As discussion in the caption of the figure, the result supports our claim that catastrophic forgetting happens when the data across clients are non-i.i.d.
>
> 2. We are sorry for the misunderstanding. We named the experiment as CIFAR-10 (uniform) since each client has only five random images sampled from all classes following a uniform distribution, as detailed in Section 4.1. However, the data is not homogeneously distributed across the clients.
>
> 3. The rationale behind the use of synthetic (or pseudo) data is mentioned in the discussion immediately following Assumption 2 and explained in more detail in Section 3.1, and its functions (i.e., how it actually helps alleviate forgetting in local training) are described in Section 3.2. Our experimental results in Section 4.3 demonstrate that the pseudo data does help with catastrophic forgetting in practice. Briefly, with Assumption (2), we assume that changes in the predicted values on previous training data at the other clients can be limited by constraining the predicted values on data near the local data at the current client. Hence, a set of pseudo data is generated to be close to the local data. The locally trained parameters are then regularized by the pseudo data to make sure that the predicted values on the pseudo data is almost unchanged before and after the local training stage. Since the pseudo data contains similar Fisher information as the (real) previous training data at the other clients as shown in Figure 2, any model that performs well on the pseudo data would likely perform well on the previous training data too. In other words, the pseudo data helps alleviate the forgetting issue by achieving a similar effect as the previous training data in the regularization of the parameters of the local model. We have expanded the discussion below Assumption 2 a bit to make this intuition more obvious.
>
> 4. As mentioned in the above response, Assumption 2 is behind the motivations for generating pseudo data in Section 3.1 and formulating the regularization on locally trained parameters in Section 3.2.
>
> 5. As the pseudo data $D_i^s$ are generated with the parameters $θ^{(t-1)}$, the loss on the pseudo data reaches the minimum at $θ^{(t-1)}$. As explained in the same sentence below equation 10, equation 4 should really be an equation instead of an inequality, but empirically, the inequality formulation would often make the optimization process more robust and result in non-negative values of $w_p$ and $w_s$.

---

> > ### Comment · Reviewer_9rQY · 2021-11-30
> > **Response**
> >
> > Thank you for your clarifications.
> > I found Fig 1 vs Fig C.1 give a convincing justification.
> >
> > There were a few points I misunderstood but are now clearer in this version.
> >
> > I will raise my score to 6 but I am concerned about the setting of hyperparameters on the test set another reviewer raise, so I will not raise further

---

> > > ### Author Response · Authors · 2021-12-02
> > > **Further Response:**
> > >
> > > Thanks so much for your positive response. Please see our further response to the 4th reviewer (YGd2) concerning the tuning of the hyper-parameters.

---

### Official Review · Reviewer_6gnX · 2021-11-02

**Correctness:** 3
**Technical Novelty And Significance:** 2
**Empirical Novelty And Significance:** 2
**Recommendation:** 5
**Confidence:** 2

**Main Review:**

This paper adopts a new perspective to improve federated learning, i.e., alleviating the catastrophic forgetting issue. This paper is clearly written and it is easy to follow. The proposed method is well explained. Experiments are solid with different datasets and baselines.

I have the following concerns and questions. I would really appreciate it if  the authors could address them.
1. Catastrophic forgetting issue is mainly studied in the scenario where the neural network is trained sequentially on multiple tasks, such as continual learning. However, in federated learning, we only consider one task as the joint objective for all clients. It is not clear why catastrophic forgetting could occur in federated learning as well. I expect more discussion on the motivation of this problem under FL settings, especially why your solution intuitively solves it.

2. Related to my first concern, the authors mentioned that “the locally trained models suffer from severe forgetting of the knowledge of previous training data”. Given that each selected client needs to do local training for S epochs, I am not exactly sure why previous training data could be forgotten. Is it because of random selection of clients? It would be helpful to precisely describe what “previous training data” means in FL.

3. In Figure 1, I assume that loss^(t-1) is computed using the averaged model from all selected clients. How about loss^t?  I am not sure if loss^t is only using the local model at client i or the averaged model from all selected clients.

4. The assumption (2) in Section 3 is highly related to the assumption that different clients should have similar examples (considering the perturbation). I am wondering how practical this assumption (2) is when the data is non-i.i.d. across different clients.

5. The experiments showed that the proposed method converges faster than the baseline methods in terms of the number of rounds. However, the additional cost incurred by pseudo data generation and regularization could result in longer wall-clock time per round. It would be great if the authors can compare the convergence rate in terms of wall-clock time.

6. The dataset and data split mechanism in the evaluation section is not the real-world FL dataset (see reference [1-3] as below), which is hard to empirically convince me. For example, FedReg may benefit more by alleviating the catastrophic forgetting when each client only has one or two classes. So I would like to see stronger experiments including the settings and dataset.

Reference:
[1] Kairouz, Peter, et al. "Advances and open problems in federated learning." arXiv preprint arXiv:1912.04977 (2019).
[2] Lai, Fan, et al. "FedScale: Benchmarking model and system performance of federated learning." arXiv preprint arXiv:2105.11367 (2021).
[3] Yuan, Honglin, et al. "What Do We Mean by Generalization in Federated Learning?." arXiv preprint arXiv:2110.14216 (2021)

**Summary Of The Paper:**

This paper considers the catastrophic forgetting issue in federated learning. The authors observe that this issue is (at least partially) responsible for slow convergence of existing FL methods when the data are not independently and identically distributed (non-i.i.d.) across different clients. This paper proposes FedReg, an algorithm to alleviate the catastrophic forgetting issue by regularizing the local model parameters on the generated pseudo data.

**Summary Of The Review:**

I have been working on related areas and I have read this paper carefully.

---

> ### Author Response · Authors · 2021-11-23
> **Response to Reviewer 6gnX**
>
> Thanks for carefully reading our submission and raising many insightful questions. In the following, we respond to each of your comments concerning the weakness of our manuscript in detail. For your convenience, we have also highlighted the main changes in the revised manuscript.
>
> 1. We agree with you that the catastrophic forgetting issue has been mainly studied in the sequential learning of multiple tasks. It is caused by the difference between the distributions of data concerning different tasks [3]. In federated learning with non-i.i.d clients, although the goal is to find an optimal model for the joint objective for all clients, the clients are trained separately (or locally) and the distributions of the local data are often different from the distribution of the global data. Thus, a locally trained model might forget the knowledge of previous training data, i.e., data sampled at the clients in previous rounds. This forgetting issue is vital when the data across clients are non-i.i.d, as shown in Figure 1. In contrast, when the data is homogenously distributed across clients, it is not clearly observed, as shown in supplementary Figure C.1 of the revised manuscript. We have expanded the second paragraph of the Introduction section a little bit (including a new supplementary figure) to discuss the catastrophic forgetting issue in FL. The basic ideas behind our proposed method, FedReg, are also sketched in this paragraph, while the details of the method (including why it works) are given in Sections 3.1 and 3.2.
>
> 2. Here, “previous training data” refers to the data sampled at the other clients in previous rounds. Again, knowledge contained in these data could be forgotten due to the discrepancy between local data distributions and the global data distribution. We have clarified the use of “previous training data” throughout the revised manuscript.
>
> 3. Thanks for the comment. The details of the computation can be found in Appendix B.1. In particular, $Loss^{(t)}$ is the average loss computed using the local models sent from the clients sampled in round t, as we claim that the forgetting issue happens after each local training stage. We have added a reference to Appendix B.1 in the caption of Figure 1.
>
> 4. Assumption 2 does not actually rely on the similarity between clients. We propose this assumption to constrain changes in the predicted values on data from the other clients during the local training stage. Since perturbation is applied to the pseudo data, this assumption only requires that the pseudo data is similar to the local data.
>
> 5. Thanks for the comment. Since our main objective is to reduce the communication costs in training (as mentioned in the beginning of Introduction), we discuss the efficiency of the compared FL methods in terms of the number of communication rounds. We agree with the reviewer that the extra computation will slow down the wall-clock time, and have added a supplementary Table C.2 to report the wall-clock time required in the experiments of Table 1. Theoretically, FedReg may require up to three times the local training time of FedAvg. We leave as future work how to reduce the computation costs of FedReg.
>
> 6. Thanks for the suggestion. The same suggestion was also made by Reviewer 1. We have included an experiment on a real-world dataset, Landmarks-User-160k [1, 2], and presented the result in supplementary Table C.1 of the revised manuscript. In this data, each client has images from 29 classes on the average. Due to the limitation of time and computation resources, we are only able to report the result after 350 communication rounds at this point. The experiment is still running and the result after more reasonable rounds will be updated in the final version (we estimate that it might take up to 1000 rounds for the methods to converge). We can already see that FedReg is significantly outperforming the baselines (its ACC after 350 rounds is 0.090 while the best ACC of the baselines is 0.065). Although this result is incomplete, we hope it is still helpful to demonstrate the advantage of our method on large-scale real-world datasets. However, we agree that FedReg benefits more when the number of classes at each client is far less than the total number of classes in the dataset, in which case the heterogeneity among clients will cause severe forgetting.

---

> > ### Author Response · Authors · 2021-11-23
> > **Reference**
> >
> > [1] Lai, F., Dai, Y., Zhu, X., Madhyastha, H. V., & Chowdhury, M. (2021, October). FedScale: Benchmarking model and system performance of federated learning. In Proceedings of the First Workshop on Systems Challenges in Reliable and Secure Federated Learning (pp. 1-3).
> >
> > [2] Hsu, T. M. H., Qi, H., & Brown, M. (2020). Federated visual classification with real-world data distribution. In Computer Vision–ECCV 2020: 16th European Conference, Glasgow, UK, August 23–28, 2020, Proceedings, Part X 16 (pp. 76-92). Springer International Publishing.
> >
> > [3] Buzzega, P., Boschini, M., Porrello, A., Abati, D., & Calderara, S. (2020). Dark Experience for General Continual Learning: a Strong, Simple Baseline. In 34th Conference on Neural Information Processing Systems (NeurIPS 2020).

---

> > ### Comment · Reviewer_6gnX · 2021-11-30
> > **follow-up**
> >
> > Thank the authors for addressing my concerns.
> >
> > (1) For the catastrophic forgetting motivation, thanks a lot for explaining what the previous training data means. But I guess the seriousness of this catastrophic forgetting issue depends on (1) how different the data distribution is across clients and (2) how different the set of participating clients is between two consecutive rounds. I believe the motivation would be stronger if the authors could illustrate the exact setting for getting Figure 1, e.g., how data distribution differs across clients and how the set of participating clients differs between two consecutive rounds. Ideally, a real-world FL dataset should be used (see the point 6 in my original comments) so that the readers can have a better sense how practice the motivation is.
> >
> > (2) As for the computation overhead, I think it's better to have more direct experiments to demonstrate that the overhead will not offset the benefits in terms of communication rounds. This is because ultimately we would like to reduce the total wall-clock time for training.
> >
> > I would prefer to keep my score now until the above two concerns are addressed.

---

> > > ### Author Response · Authors · 2021-12-01
> > > **Response**
> > >
> > > Thank you for your follow-up comments on our response to your initial review. In the following, we address your two specific concerns.
> > >
> > > 1.	The details of the experimental data and setting used in Figure 1(a) are given in section 4.1 and supplementary section B.2. More specifically, in these experiments, each client has images from one class, and in each round, 0.2% of the clients are randomly sampled to participate in training. In other words, the data distributions are completely different across the clients and very different clients are sampled in consecutive rounds. As mentioned in our manuscript and also previous response, we agree that the severity of the catastrophic forgetting issue depends on the heterogeneity of data across the clients. In fact (as mentioned in our previous response), the newly added supplementary Figure C.1 in the revised manuscript illustrates that when the data is homogeneous across the clients, the forgetting issue does not appear to exist. However, we do not think that the difference between the sets of participating clients in consecutive training rounds has much to do with the forgetting issue, since it is mostly caused by the difference between the distributions of the local data and the global data. That is, FL may still suffer from the forgetting issue even if all clients are sampled in each round. Hence, we wonder if this particular question/suggestion is due to some misunderstanding?
> > >
> > >       As for your suggestion of motivating the problem using a real-world dataset, we observe that real-world datasets are often heterogeneous enough to cause such forgetting. For instance, in the real-world dataset that we have added to the revised manuscript (based on your previous request), Landmarks-User-160k, there are a total of 2028 classes but each client has images from only 29 classes. Hence, the clients have very differently distributed data, and our method is able to take advantage of this and significantly outperform the baseline methods, as shown in supplementary Table C.1 of the revised manuscript.
> > >
> > >
> > > 2.	As stated in the literature [1, 2] and mentioned in our manuscript and previous response, we consider how to reduce communication costs as the main challenge in FL and hence measure the performance of a method mainly in terms of its communication cost. Although we agree with the reviewer that the overall training time is important, especially because our method requires more time than some of the baselines per training round, we observe that it is difficult to translate communication costs (rounds) into time in simulation studies. For example, one would have to make realistic assumptions about parameters such as bandwidth, network traffic, etc. Perhaps, for this reason, none of the papers corresponding to the baseline methods used in our experiments discussed performance in terms of total training time.
> > >
> > > [1] Karimireddy, S. P., Kale, S., Mohri, M., Reddi, S., Stich, S., & Suresh, A. T. (2020, November). Scaffold: Stochastic controlled averaging for federated learning. In International Conference on Machine Learning (pp. 5132-5143). PMLR.
> > > [2] Konečný, J., McMahan, H. B., Yu, F. X., Richtárik, P., Suresh, A. T., & Bacon, D. (2016). Federated learning: Strategies for improving communication efficiency. arXiv preprint arXiv:1610.05492.

---

> > > > ### Comment · Reviewer_6gnX · 2021-12-01
> > > > **more about the catastrophic forgetting motivation**
> > > >
> > > > Thank the authors for the reply.
> > > >
> > > > Especially, the authors stated that "That is, FL may still suffer from the forgetting issue even if all clients are sampled in each round." I am not sure if I can understand why this statement is true, which further confuses me about the catastrophic forgetting motivation.
> > > >
> > > > Let us consider the case where all the clients are participating at every round. In your previous response, you mentioned that both $loss^{t-1}$ and $loss^t$ are averaged loss using local models from all participating clients. If all the clients are participating at every round, then both $loss^{t-1}$ and $loss^t$ are averaged over all clients. This "averaging" operation is supposed to take into account the local data distributions of all the clients (the local distribution probably differs from client to client), which should be able to decrease the average loss $loss^t$ when the learning rate is properly chosen. To put it another way, if Figure 1 is true for every round, then it implies that $loss^t$ will keep increasing as training goes. Does it mean that classical FL methods like FedAvg will not converge?
> > > >
> > > > I am not sure if there is any misunderstanding. But I would appreciate it if the authors could further clarify the above concern.

---

> > > > > ### Author Response · Authors · 2021-12-02
> > > > > **Further Response**
> > > > >
> > > > > Thanks so much for your prompt follow-up on our last response. There is indeed a misunderstanding of the notations. As explained in the caption of Figure 1, $loss^{(t-1)}$ denotes the average loss on each piece of local data at the end of round $t-1$ and $loss^{(t)}$ denotes the average loss on each piece of local data at the end of the local training stage in round $t$ (but before its aggregation stage). In other words, the former is computed using the global model after round $t-1$ and the latter is computed using the local models trained in round $t$, as clearly described in supplementary section B.1. This is intentional because the goal of our work is to relieve the forgetting issue in local training (as stated in the title of the manuscript). Classical FL methods such as FedAvg usually converge because the aggregated (global) model improves after each round.

---

> > > > > > ### Comment · Reviewer_6gnX · 2021-12-04
> > > > > > **still concerned**
> > > > > >
> > > > > > Thanks for further clarifying the notations $loss^t$ and $loss^{t-1}$. I think the motivation would be more clear if their definitions are explicitly provided in the main text where Figure 1a is discussed.
> > > > > >
> > > > > > What's more important, I think it is more common to use the aggregated loss at the end of each round to keep track of the training progress for federated learning, because the central server maintains only the aggregated model. As I have pointed out in my last comment, the aggregation stage should be able to take into account the local data distributions of all the clients (the local distribution probably differs from client to client), thus decreasing the aggregated loss when the learning rate is properly chosen. As long as the aggregated loss keeps decreasing, I am not sure whether catastrophic forgetting is still a serious issue. In other words, catastrophic forgetting issue illustrated by the pre-aggregation loss (such as $loss^t$ and $loss^{t-1}$) might be some kind of transient issue that would simply disappear after the aggregation stage is done.
> > > > > >
> > > > > > Having said above, it might be better to use the aggregated loss for illustrating the catastrophic forgetting issue if the issue still exists.

---

> > > > > > > ### Author Response · Authors · 2021-12-05
> > > > > > > **Further Response:**
> > > > > > >
> > > > > > > Thanks so much once again for your follow-up. Since the notations $loss^{(t)}$ and $loss^{(t-1)}$ are only used in Figure 1 and supplementary section B.1, and they are clearly defined in these places, we are not sure if it would help to repeat them in the main text. Note that we talk about the forgetting issue explicitly in the context of the local training stage throughout the main text without referring to these technical notations, e.g., “the catastrophic forgetting issue during the local training stage on each individual client” in the abstract, “the locally trained models suffer from severe forgetting of the knowledge of previous training data at the other clients” in the introduction section. So, it should be clear to the reader that we are referring to the knowledge learned by the local models rather than the global model when discussing this issue even if he/she decides to skip the technical details in Figure 1 or supplementary section B.1.
> > > > > > >
> > > > > > > While we agree that aggregated losses should be used to track the overall progress in FL and the aggregation stage may alleviate the forgetting issue observed in the local training stage to some extent, we disagree that the forgetting issue in local training is not a worthy research topic, even if it is “transient”. In fact, our experiments have demonstrated that by addressing the issue directly and timely using pseudo and perturbed data in every local training stage, the communication costs of FL can be significantly reduced, perhaps because the aggregation step of FL is not designed specifically to address this particular issue.

---

### Official Review · Reviewer_UTZk · 2021-11-05

**Correctness:** 4
**Technical Novelty And Significance:** 3
**Empirical Novelty And Significance:** Not applicable
**Recommendation:** 6
**Confidence:** 3

**Main Review:**

Strength:
(1) Very interesting and novel idea: it is the first paper I have read about showing adversarial examples of global model can help to prevent forgetting in local models. Previous works are mainly focused on directly regularizing the parameters or training on previous data samples.
(2) good result comparing with other methods.

Weakness:
(1) writing is a little bit unclear in describing the method. Clearer definition of notations in section 3.1 and 3.2 will be really helpful.
(2) experiments only focus on small scale dataset with small number of classes. It will be really helpful to include larger scale dataset with natural non-iid data distribution.

**Summary Of The Paper:**

This paper aims to alleviate forgetting in federated learning on non iid data. The method proposed FedReg focuses on regularizing locally trained parameters with the loss on generated pseudo data, which are based on adversarial examples of the global model in the previous step and the adversarial examples of the local model at current step. This paper experiments on several small scale dataset with large number of clients. It shows improved performance and less forgetting during the training.

**Summary Of The Review:**

Because I think the main idea of the paper is really interesting, I will give 6-7 grade. I am more than willing to increase my score if more large scale experiments are added and writing is improved.

---

> ### Author Response · Authors · 2021-11-23
> **Response to Reviewer UTZk**
>
> Thanks so much for your careful review and encouraging comments. In the following, we give our detailed response to your comments concerning the weakness of our submission.
>
> 1. We have improved the presentation of section 3 by adding definitions and explanations of several notations such as $d^-$, $D_i^s$, $D_i^p$, etc. and clarifying unclear phrases such as “previous training data", etc. Moreover, we have reorganized the steps outlined in equations (2) and (3) so the procedures for generating pseudo and perturbed data are easier to follow. We hope that these changes have made the section more readable although it is still packed with mathematical notations and equations (which we think are necessary for the rigor and integrity of the discussion). For your convenience, the main changes are highlighted in the revised manuscript.
>
> 2. Thanks for this suggestion. We have included an experiment on a real-world dataset, Landmarks-User-160k [1, 2], which consists of 2,028 classes from 1,262 users. The results are added as supplementary Table C.1 of the revised manuscript. Due to the limitation of time and computation resources, we are only able to report the result after 350 communication rounds at this point. The experiment is still running and the result after more reasonable rounds will be updated in the final version (we estimate that it might take up to 1000 rounds for the methods to converge). We can already see that FedReg is significantly outperforming the baselines (its ACC after 350 rounds is 0.090 while the best ACC of the baselines is 0.065). Although this result is incomplete, we hope it is still helpful to demonstrate the advantage of our method on large-scale real-world datasets.
>
> [1] Lai, F., Dai, Y., Zhu, X., Madhyastha, H. V., & Chowdhury, M. (2021, October). FedScale: Benchmarking model and system performance of federated learning. In Proceedings of the First Workshop on Systems Challenges in Reliable and Secure Federated Learning (pp. 1-3).
>
> [2] Hsu, T. M. H., Qi, H., & Brown, M. (2020). Federated visual classification with real-world data distribution. In Computer Vision–ECCV 2020: 16th European Conference, Glasgow, UK, August 23–28, 2020, Proceedings, Part X 16 (pp. 76-92). Springer International Publishing.

---

> > ### Comment · Reviewer_UTZk · 2021-12-03
> > **response**
> >
> > Thank you for the reply! And really sorry for the late response due to some personal issues! I think the paper is clearer now especially the notations. And I truly appreciate the additional experiments. As I said I really like the main idea of the paper. But as other reviewer points out on the issue with hyper-parameter tunning I didn't notice, I am afraid I cannot further raise my score.

---

### Decision · Program_Chairs · 2022-01-20

**Decision:**

Accept (Poster)

**Comment:**

The paper considers the problem of distributed optimization in the Federated Learning setting in particular when the data in the clients is non-i.i.d. The paper points to the problem of catastrophic forgetting during the local update stages to be a cause for the bad training of models and proposes to fix it via introducing a pseudo loss, which are based on adversarial examples of the global model in the previous step and the adversarial examples of the local model at current step.

The paper's core idea was generally appreciated by the reviewers as well as the favorable evaluation of the proposed method when compared with other existing algorithms. The authors also provided further additional information during the rebuttal period and addressed author comments adding enough justification to the paper to resolve issues in the submission. One lingering issue that remains is the hyper-parameter tuning on test set results that was performed. The authors have promised to redo experiments based on validation sets.

Overall the paper is borderline but I am recommending accept based on novelty of the idea and strong experimental results.